# Squared Neural Families:
# A New Class of Tractable Density Models

**Russell Tsuchida**[*]
Data61-CSIRO

**Cheng Soon Ong**
Data61-CSIRO &
Australian National University

**Dino Sejdinovic**
School of CMS & AIML,
The University of Adelaide

## Abstract

Flexible models for probability distributions are an essential ingredient in many machine learning tasks. We develop and investigate a new class of probability distributions, which we call a Squared Neural Family (SNEFY), formed by squaring the 2-norm of a neural network and normalising it with respect to a base measure. Following the reasoning similar to the well established connections between infinitely wide neural networks and Gaussian processes, we show that SNEFYs admit closed form normalising constants in many cases of interest, thereby resulting in flexible yet fully tractable density models. SNEFYs strictly generalise classical exponential families, are closed under conditioning, and have tractable marginal distributions. Their utility is illustrated on a variety of density estimation, conditional density estimation, and density estimation with missing data tasks.

## 1 Introduction

Probabilistic modelling lies at the heart of machine learning. In both traditional and contemporary settings, ensuring the probability model is appropriately normalised (or otherwise bypassing the need to compute normalising constants) is of central interest for maximum likelihood estimation and related statistical inference procedures. Tractable normalising constants allow the use of probability models in a variety of applications, such as anomaly detection, denoising and generative modelling.

Traditional statistical approaches [27, 5] rely on mathematically convenient models such as exponential families [50]. Such models can often be normalised in closed form, but are often only suitable for relatively low-dimensional and simple data. Early approaches in deep learning use neural networks as energy functions inside Gibbs distributions [21, Equation 2]. Such distributions typically have very intractable normalising constants, and so either surrogate losses for the negative log likelihood involving the energy function are used or MCMC, score matching [16], variational or sampling methods [15, 30] are used to approximate the normalising constant. See [35, §2.6.1, §2.2] for more energy-based and other methods.

Modern computer technology and power allows us more flexible models [8, page 68], by partially or completely relaxing the requirement for mathematical tractability. Contemporary high-dimensional modelling, such as generative image models, rely primarily on neural network models [18, 11, 35, 34]. For example, normalising flows [40] use constrained neural network layers to transform random variables from base measures with tractable Jacobian determinants associated with appropriately constrained but flexible pushforward maps. Similarly to normalising flows, we use neural networks to define expressive densities, but we model the densities directly without using constrained transformations of random variables. Moreover, our approach can be readily applied in conjunction with normalising flows and other deep learning devices such as deep feature extractors.

---

[*]Software available at `https://github.com/RussellTsuchida/snefy`.

37th Conference on Neural Information Processing Systems (NeurIPS 2023).

| Support $\mathbb{X}$ | Base measure $\mu$ | Sufficient statistic $\boldsymbol{t}(\boldsymbol{x})$ | Activation function $\sigma$ | $\boldsymbol{b} \neq \boldsymbol{0}$ | Kernel $k_{\sigma,\boldsymbol{t},\mu}$ |
|---|---|---|---|---|---|
| **Any setting mirroring a previously derived NNGPK, e.g.** | | | | | |
| $\mathbb{R}^d$ | $\Phi_{\boldsymbol{C},\boldsymbol{0}}$ | $\mathrm{Id}(\boldsymbol{x})$ | erf | ✗ | [52] |
| | | | $(\cdot)_+^p, p \in \mathbb{N}$ | ✗ | [4] |
| | | | LReLU | ✗ | [48] |
| | | | GELU [14] | ✗ | [47] |
| | | | cos | ✗ | [37] |
| **Any tractable exponential family [50, 32] setting, e.g.** | | | | | |
| $\mathbb{R}^d$ | $\Phi_{\boldsymbol{C},\boldsymbol{m}}$ | $\frac{\boldsymbol{x}}{\|\boldsymbol{x}\|_2}$ | exp | ✓ | Kernel 3 |
| $\mathbb{S}^{d-1}$ | Uniform | $\mathrm{Id}(\boldsymbol{x})$ | | ✓ | |
| $\mathbb{R}^d$ | $\Phi_{\boldsymbol{C},\boldsymbol{m}}$ | | | ✓ | Kernel 7 |
| $\{0, 1, 2, \ldots\}$ | $(x!)^{-1}\nu$ | | | ✓ | Kernel 8 |
| **New tractable integration settings, e.g.** | | | | | |
| $\mathbb{R}^d$ | $\Phi_{\boldsymbol{C},\boldsymbol{m}}$ | $\mathrm{Id}(\boldsymbol{x})$ | cos | ✓ | Kernel 2 |
| $[0, 1]^d$ | Uniform | $\Phi^{-1}(\boldsymbol{x})$ | | | |
| $\mathbb{R}^d$ | $\Phi_{\boldsymbol{C},\boldsymbol{m}}$ | $\mathrm{Id}(\boldsymbol{x})$ | $\mathrm{Snake}_a$ [55, 39] | ✓ | Kernel 6 |

Table 1: Examples of settings admitting a closed-form for the normalising constant $\mathrm{z}(\boldsymbol{V}, \boldsymbol{\Theta})$ (2) by leveraging a closed-form NNK $k_{\sigma,\boldsymbol{t},\mu}$ (4). In each case, $\mathrm{z}(\boldsymbol{V}, \boldsymbol{\Theta}) = \mathrm{Tr}(\boldsymbol{V}^\top \boldsymbol{V} \boldsymbol{K}_{\boldsymbol{\Theta}})$, where the entries of the matrix $\boldsymbol{K}_{\boldsymbol{\Theta}}$ are described according to the NNK $k_{\sigma,\boldsymbol{t},\mu}$ in Identity 1. $\Phi_{\boldsymbol{C},\boldsymbol{m}}$ denotes the CDF of a multivariate normal distribution with mean $\boldsymbol{m}$ and covariance matrix $\boldsymbol{C}$ and $\nu$ denotes counting measure. Rows with citations have been considered previously in the context of NNGPKs, but not as normalising constants and not with a reversal of the role of input and parameter. Note the cases where $\boldsymbol{b} \neq \boldsymbol{0}$; this setting is not considered by others, because when the role of parameters and data is in the usual setting, $\boldsymbol{b} = \boldsymbol{0}$ covers a sufficiently general setting. Noticing that SNEFYs strictly generalise exponential family mixture models (see § 3.2), fixing $\sigma(\cdot) = \exp(\cdot/2)$ and a given base measure $\mu$ and sufficient statistic $\boldsymbol{t}$ for which the exponential family log-partition function is known also leads to tractable normalising constants. More known closed-form kernels as well as approximate kernels that can be adapted to our setting are given in [12].

**Contributions.**   Let $(\Omega, \mathcal{F}, \mu)$ denote a measure space with $\Omega \subseteq \mathbb{R}^d$, sigma algebra $\mathcal{F}$, and nonnegative measure $\mu$. Let $\boldsymbol{V} \in \mathbb{R}^{m \times n}$ be the readout parameters of a neural network and let $\boldsymbol{W} \in \mathbb{R}^{n \times D}$ and $\boldsymbol{b} \in \mathbb{R}^n$ be the weights and biases of a hidden layer of a neural network $\boldsymbol{f} : \mathbb{R}^D \to \mathbb{R}^m$ with activation function $\sigma$. For some support $\mathbb{X} \in \mathcal{F}$, define a probability distribution $P$ (and corresponding probability density $p$ with respect to base measure $\mu$) to be proportional to squared 2-norm of the evaluation of the neural network $\boldsymbol{f}$,

$$P(d\boldsymbol{x}; \boldsymbol{V}, \boldsymbol{\Theta}) \triangleq \frac{\mu(d\boldsymbol{x})}{\mathrm{z}(\boldsymbol{V}, \boldsymbol{\Theta})} \big\| \boldsymbol{f}\left(\boldsymbol{t}(\boldsymbol{x}); \boldsymbol{V}, \boldsymbol{\Theta}\right) \big\|_2^2, \quad \boldsymbol{f}(\boldsymbol{t}; \boldsymbol{V}, \boldsymbol{\Theta}) = \boldsymbol{V}\sigma(\boldsymbol{W}\boldsymbol{t} + \boldsymbol{b}), \ \boldsymbol{\Theta} = (\boldsymbol{W}, \boldsymbol{b}), \ (1)$$

whenever the normalising constant $\mathrm{z}(\boldsymbol{V}, \boldsymbol{\Theta}) \triangleq \int_{\mathbb{X}} \big\| \boldsymbol{f}\left(\boldsymbol{t}(\boldsymbol{x}); \boldsymbol{V}, \boldsymbol{\Theta}\right) \big\|_2^2 \mu(d\boldsymbol{x})$ is finite and non-zero. Here we call $\mu$ the base measure, $\boldsymbol{t} : \mathbb{X} \to \mathbb{R}^D$ the sufficient statistic[2] and $\sigma$ the activation function. We will call the corresponding family of probability distributions, parametrised by $(\boldsymbol{V}, \boldsymbol{\Theta})$, a *squared neural family* (SNEFY) on $\mathbb{X}$, and denote it by $\mathrm{SNEFY}_{\mathbb{X},\boldsymbol{t},\sigma,\mu}$. SNEFYs are new flexible probability distribution models that strike a good balance between mathematical tractability, expressivity and computational efficiency. When a random vector $\mathbf{x}$ follows a SNEFY distribution indexed by parameters $(\boldsymbol{V}, \boldsymbol{\Theta})$, we write $\mathbf{x} \sim \mathrm{SNEFY}_{\mathbb{X},\boldsymbol{t},\sigma,\mu}(\boldsymbol{V}, \boldsymbol{\Theta})$ or simply $\mathbf{x} \sim P(\cdot; \boldsymbol{V}, \boldsymbol{\Theta})$, where there is no ambiguity.

Our main technical challenge is in exactly computing the normalising constant, $\mathrm{z}(\boldsymbol{V}, \boldsymbol{\Theta})$, where

$$\mathrm{z}(\boldsymbol{V}, \boldsymbol{\Theta}) \triangleq \int_{\mathbb{X}} \big\| \boldsymbol{f}\left(\boldsymbol{t}(\boldsymbol{x}); \boldsymbol{V}, \boldsymbol{\Theta}\right) \big\|_2^2 \mu(d\boldsymbol{x}), \quad \boldsymbol{f}(\boldsymbol{t}; \boldsymbol{V}, \boldsymbol{\Theta}) = \boldsymbol{V}\sigma(\boldsymbol{W}\boldsymbol{t} + \boldsymbol{b}), \quad \boldsymbol{\Theta} = (\boldsymbol{W}, \boldsymbol{b}). \quad (2)$$

The normalising constants we consider are special cases in the sense that they apply to specific (but commonly appearing in applications) choices of activation function $\sigma$, sufficient statistic $t$ and base measure $\mu$ over support $\mathbb{X}$. See Table 1. Our analysis both exploits and informs a connection with

---

[2]We later verify that $\boldsymbol{t}$ is indeed a sufficient statistic, see (7). Note $\boldsymbol{t}$ maps from $d$ to $D$ dimensions.

so-called neural network Gaussian process kernels (NNGPKs) [31] in a generalised form, which we refer to as neural network kernels (NNKs).

We discuss some important theoretical properties of SNEFY such as exact normalising constant calculation, marginal distributions, conditional distributions, and connections with other probability models. We then consider a deep learning setting, where SNEFYs can either be used as base distributions in (non-volume preserving) normalising flows [46], or may describe flexible conditional density models with deep learning feature extractors. We demonstrate SNEFY on a variety of datasets.

## 1.1 Background

**Notation** We use lower case non-bold (like $a$) to denote scalars, lower case bold to denote vectors (like $\boldsymbol{a}$) and upper case bold to denote matrices (like $\boldsymbol{A}$). Random variables (scalar or vector) are additionally typeset in sans-serif (like a and **a**). The special zero vector $(0, \dots, 0)$ and identity matrix elements are $\boldsymbol{0}$ and $\boldsymbol{I}$. We use subscripts to extract (groups of) indices, so that for example, $\boldsymbol{w}_i$ is the $i$th row of the matrix $\boldsymbol{W}$ (as a column vector), $\boldsymbol{v}_{\cdot,i}$ is the $i$th column of the matrix $\boldsymbol{V}$, and $b_i$ is the $i$th element of the vector $\boldsymbol{b}$. We use $\boldsymbol{\Theta} = (\boldsymbol{W}, \boldsymbol{b}) \in \mathbb{R}^{n \times (D+1)}$ to denote the concatenated hidden layer weights and biases. Correspondingly, we write $\boldsymbol{\theta}_i = (\boldsymbol{w}_i, b_i) \in \mathbb{R}^{D+1}$ for the $i$th row of $\boldsymbol{\Theta}$. We will use a number of special functions. $\Phi_{C,\boldsymbol{m}}$ denotes the cumulative distribution function (CDF) of a Gaussian random vector with mean $\boldsymbol{m}$ and covariance matrix $\boldsymbol{C}$. We also use a shorthand $\Phi_C = \Phi_{C,\boldsymbol{0}}$ and $\Phi = \Phi_{\boldsymbol{I},\boldsymbol{0}}$.

**Single hidden layer neural networks** We consider a feedforward neural network $\boldsymbol{f} : \mathbb{R}^D \to \mathbb{R}^m$,

$$\boldsymbol{f}(\boldsymbol{t}; \boldsymbol{V}, \boldsymbol{\Theta}) = \boldsymbol{V}\sigma(\boldsymbol{W}\boldsymbol{t} + \boldsymbol{b}) \tag{3}$$

with activation function $\sigma$, hidden weights $\boldsymbol{W} \in \mathbb{R}^{n \times D}$ and biases $\boldsymbol{b} \in \mathbb{R}^n$ and readout parameters $\boldsymbol{V} \in \mathbb{R}^{m \times n}$. Here $\sigma$ is applied element-wise to its vector inputs, returning a vector of the same shape.

**Neural network kernels** In certain theories of deep learning, one often encounters a bivariate Gaussian integral called the *neural network Gaussian process kernel* NNGPK. The NNGPK first arose as the covariance function of a well-behaved single layer neural network with random weights [31]. In the limit as the width of the network grows to infinity, the neural network (3) with suitably well-behaved (say, independent Gaussian) random weights converges to a zero-mean Gaussian process, so that the NNGPK characterises the law of the neural network predictions. These limiting models can be used as functional priors in a classical Gaussian process sense [53].

In our setting, the positive semidefinite (PSD) NNGPK appears in an entirely novel context, where the role of the hidden weights and biases $\boldsymbol{\theta}_i = (\boldsymbol{w}_i, b_i)$ and the data $\boldsymbol{x}$ is reversed. Instead of marginalising out the parameters and evaluating at the data, we marginalise out the data and evaluate at the parameters. The NNGPK $k_{\sigma, \mathrm{Id}, \Phi_{\boldsymbol{I}}}$ admits a representation of the form

$$k_{\sigma, \mathrm{Id}, \Phi_{\boldsymbol{I}}}(\boldsymbol{\theta}_i, \boldsymbol{\theta}_j) \triangleq \mathbb{E}_{\mathsf{x}}\big[\sigma\big(\boldsymbol{w}_i^\top \mathsf{x} + b_i\big)\sigma\big(\boldsymbol{w}_j^\top \mathsf{x} + b_j\big)\big], \quad \mathsf{x} \sim \mathcal{N}\big(\boldsymbol{0}, \boldsymbol{I}\big). \tag{4}$$

We do not discuss in detail how it is constructed in earlier works [31, 22, 33, 17], where usually $b_i = b_j = 0$[3], but not always[49]. When $b_i = b_j = 0$, closed-form expressions for the NNGPK are available for different choices of $\sigma$ and $\boldsymbol{t}$ [52, 20, 4, 48, 37, 47, 28, 13]. However, the setting of $\boldsymbol{b} \neq \boldsymbol{0}$ is important in our context (as we show in §2.2) and presents additional analytical challenges.

We will require a more general notion of an NNGPK which we call a *neural network kernel* (NNK). We introduce a function $\boldsymbol{t}$ which may be thought of as a warping function applied to the input data. Such warping is common in kernels and covariance functions and can be used to induce desirable analytical and practical properties [37, 29, 24, §5.4.3]. We also integrate with respect to more general measures $\mu$ instead of the standard Gaussian CDF, $\Phi$. We define the NNK to be

$$k_{\sigma, \boldsymbol{t}, \mu}(\boldsymbol{\theta}_i, \boldsymbol{\theta}_j) \triangleq \int_{\mathbb{X}} \sigma\big(\boldsymbol{w}_i^\top \boldsymbol{t}(\boldsymbol{x}) + b_i\big)\sigma\big(\boldsymbol{w}_j^\top \boldsymbol{t}(\boldsymbol{x}) + b_j\big)\mu(d\boldsymbol{x}). \tag{5}$$

---

[3]After accounting for the reversal of parameters and data.

## 2 Closed form squared neural families

### 2.1 Normalising constants

Observe from (2) that by swapping the order of integration and multiplication by $\boldsymbol{V}$, the normalising constant is quadratic in elements of $\boldsymbol{V}$. The coefficients of the quadratic depend on $\boldsymbol{\Theta} = (\boldsymbol{W}, \boldsymbol{b})$. We now characterise these coefficients of the quadratic in terms of the NNK evaluated at rows $\boldsymbol{\theta}_i, \boldsymbol{\theta}_j$ of $\boldsymbol{\Theta}$, which are totally independent of $\boldsymbol{V}$. The proof of the following is given in Appendix A.

**Identity 1.** *The integral* (2) *admits a representation of the form*

$$\mathrm{z}(\boldsymbol{V}, \boldsymbol{\Theta}) = \mathrm{Tr}\left(\boldsymbol{V}^\top \boldsymbol{V} \boldsymbol{K}_{\boldsymbol{\Theta}}\right) \tag{6}$$

*where* $k_{\sigma, \boldsymbol{t}, \mu}$ *is as defined in* (5)*, and* $\boldsymbol{K}_{\boldsymbol{\Theta}}$ *is the PSD matrix whose* $ij$*th entry is* $k_{\sigma, \boldsymbol{t}, \mu}(\boldsymbol{\theta}_i, \boldsymbol{\theta}_j)$.

By Identity 1, the normalised measure (1) then admits the explicit representation

$$P(d\boldsymbol{x}; \boldsymbol{V}, \boldsymbol{\Theta}) = \frac{\mathrm{Tr}\left(\boldsymbol{V}^\top \boldsymbol{V} \widetilde{\boldsymbol{K}_{\boldsymbol{\Theta}}}(\boldsymbol{x})\right)}{\mathrm{Tr}\left(\boldsymbol{V}^\top \boldsymbol{V} \boldsymbol{K}_{\boldsymbol{\Theta}}\right)} \mu(d\boldsymbol{x}) = \frac{\mathrm{vec}(\boldsymbol{V}^\top \boldsymbol{V})^\top \mathrm{vec}(\widetilde{\boldsymbol{K}_{\boldsymbol{\Theta}}}(\boldsymbol{x}))}{\mathrm{vec}(\boldsymbol{V}^\top \boldsymbol{V})^\top \mathrm{vec}(\boldsymbol{K}_{\boldsymbol{\Theta}})} \mu(d\boldsymbol{x}),$$

where $\widetilde{\boldsymbol{K}_{\boldsymbol{\Theta}}}(\boldsymbol{x})$ is the PSD matrix whose $ij$th entry is $\sigma(\boldsymbol{w}_i^\top \boldsymbol{t}(\boldsymbol{x}) + b_i)\sigma(\boldsymbol{w}_j^\top \boldsymbol{t}(\boldsymbol{x}) + b_j)^\top$. We used the cyclic property of the trace, writing the numerator as $\mathrm{Tr}\left(\boldsymbol{\sigma}^\top \boldsymbol{V}^\top \boldsymbol{V} \boldsymbol{\sigma}\right) = \mathrm{Tr}\left(\boldsymbol{V} \boldsymbol{\sigma} \boldsymbol{\sigma}^\top \boldsymbol{V}^\top\right) = \mathrm{Tr}\left(\boldsymbol{V}^\top \boldsymbol{V} \boldsymbol{\sigma} \boldsymbol{\sigma}^\top\right)$. We emphasise again that the role of the data $\boldsymbol{t}(\boldsymbol{x})$ and the hidden weights and biases $\boldsymbol{\Theta}$ in the NNK $k_{\sigma, \boldsymbol{t}, \mu}$ are reversed compared with how they have appeared in previous settings. We may compute evaluations of $k_{\sigma, \boldsymbol{t}, \mu}$ in closed form for special cases of $(\sigma, \boldsymbol{t}, \mu)$ using various identities in $\mathcal{O}(d)$, where $d$ is the dimensionality of the domain of integration, as we soon detail in § 2.2. Combined with the trace inner product, this leads to a total cost of computing $\mathrm{z}(\boldsymbol{V}, \boldsymbol{\Theta})$ of $\mathcal{O}(m^2 n + dn^2)$, where $n$ and $m$ are respectively the number of neurons in the first and second layers.

**Remark 1** (Alternative parameterisations)**.** *`SNEFY` models depend on readout parameters* $\boldsymbol{V}$ *only through the direction of* $\mathrm{vec}(\boldsymbol{V}^\top \boldsymbol{V})$ *and not on its norm or sign. For example, one can always find another parameterisation of readout parameters that results in the same probability distribution but has a normalising constant of* $1$*. Furthermore, noticing that* $\boldsymbol{V}$ *only appears as a PSD matrix* $\boldsymbol{M} \triangleq \boldsymbol{V}^\top \boldsymbol{V}$ *of rank at most* $\min(m, n)$*, one may alternatively parameterise a* `SNEFY` *by* $(\boldsymbol{M}, \boldsymbol{\Theta})$.

### 2.2 Neural network kernels

In § 2.1, we reduced computation of the integral (2) to computation of a quadratic involving evaluations of the NNK (5). Several closed-forms are known for different settings of $\sigma$, all with $\boldsymbol{t} = \mathrm{Id}$ and $\boldsymbol{b} = \boldsymbol{0}$. The motivation behind derivation of existing known results is from the perspective of inference in infinitely wide Bayesian neural networks [31] or to derive certain integrals involved in computing predictors of infinitely wide neural networks trained using gradient descent [17]. Here we describe some new settings that have not been investigated previously that are useful for the new setting of `SNEFY`. Recall from (5), that our kernel, $k_{\sigma, \boldsymbol{t}, \mu}$, is parametrised by activation function $\sigma$, warping function (sufficient statistic) $\boldsymbol{t}$, and base measure $\mu$. All derivations are given in Appendix B.

The first kernel describes how we may express the kernels with arbitrary Gaussian base measures $\Phi_{\boldsymbol{C}, \boldsymbol{m}}$ in terms of the kernels with isotropic Gaussian base measures $\Phi$. This means that it suffices to consider isotropic Gaussian base measures in place of arbitrary Gaussian base measures.

**Kernel 1.** $k_{\sigma, \mathrm{Id}, \Phi_{\boldsymbol{C}, \boldsymbol{m}}}(\boldsymbol{\theta}_i, \boldsymbol{\theta}_j) = k_{\sigma, \mathrm{Id}, \Phi}(\mathcal{T}\boldsymbol{\theta}_i, \mathcal{T}\boldsymbol{\theta}_j)$, *where* $\mathcal{T}\boldsymbol{\Theta} = (\boldsymbol{W}\boldsymbol{A}, \boldsymbol{b} + \boldsymbol{W}\boldsymbol{m})$, $\mathcal{T}\boldsymbol{\theta}_i = (\boldsymbol{w}_i^\top \boldsymbol{A}, b_i + \boldsymbol{w}_i^\top \boldsymbol{m})$ *and* $\boldsymbol{A}$ *is a matrix factor such that covariance* $\boldsymbol{C} = \boldsymbol{A}\boldsymbol{A}^\top$.

This kernel can also be used to describe kernels corresponding with Gaussian mixture model base measures. The second kernel we describe is a minor extension to the case $\boldsymbol{b} \neq \boldsymbol{0}$ of a previously considered kernel [37].

**Kernel 2.** $k_{\cos, \mathrm{Id}, \Phi}(\boldsymbol{\theta}_i, \boldsymbol{\theta}_j) = \frac{\cos|b_i - b_j|}{2} \exp\left(\frac{-\|\boldsymbol{w}_j - \boldsymbol{w}_j\|^2}{2}\right) + \frac{\cos|b_i + b_j|}{2} \exp\left(\frac{-\|\boldsymbol{w}_j + \boldsymbol{w}_j\|^2}{2}\right)$.

A similar result and derivation holds for the case of $k_{\sin, \mathrm{Id}, \Phi}$, which we do not reproduce here. We now mention a case that shares a connection with exponential families (see § 3.2 for a detailed description of this connection). The following and more $\sigma = \exp$ cases are derived in Appendix C.

**Kernel 3.** *Define $proj_{\mathbb{S}^{d-1}}(\boldsymbol{x}) \triangleq \boldsymbol{x}/\|\boldsymbol{x}\|$ to be the projection onto the unit sphere. Then*

$$k_{\exp,proj_{\mathbb{S}^{d-1}},\Phi}(\boldsymbol{\theta}_i, \boldsymbol{\theta}_j) = \exp(b_i + b_j) \frac{\Gamma(d/2)2^{d/2-1}I_{d/2-1}\big(\|\boldsymbol{w}_i + \boldsymbol{w}_j\|\big)}{\|\boldsymbol{w}_i + \boldsymbol{w}_j\|^{d/2-1}},$$

*where $I_p$ is the modified Bessel function of the first kind of order $p$. In the special case $d = 3$, we have the closed-form $k_{\exp,proj_{\mathbb{S}^2},\Phi}(\boldsymbol{\theta}_i, \boldsymbol{\theta}_j) = \exp(b_i + b_j) \frac{\big(e^{\|\boldsymbol{w}_i + \boldsymbol{w}_j\|} - e^{-\|\boldsymbol{w}_i + \boldsymbol{w}_j\|}\big)}{2\|\boldsymbol{w}_i + \boldsymbol{w}_j\|}.$*

We end this section with a new analysis of the $\mathrm{Snake}_a$ activation function, given by

$$\mathrm{Snake}_a(z) = z + \frac{1}{a}\sin^2(az) = z - \frac{1}{2a}\cos(2az) + \frac{1}{2a}.$$

The $\mathrm{Snake}_a$ function [55] is a neural network activation function that can resemble the ReLU on an interval for special choices of $a$, is easy to differentiate, and as we see shortly, admits certain attractive analytical tractability. We note that a similar activation function has been found using reinforcement learning to search for good activation functions [39, Table 1 and 2, row 3], up to an offset and hyperparameter $a = 1$. The required kernel is expressed in terms of the linear kernel (Kernel 4) and the kernel corresponding with the activation function of [39], i.e. snake without the offset, $\mathrm{Snake}_a(\cdot) - \frac{1}{2a}$ (Kernel 5). We first describe the linear kernel.

**Kernel 4.** $k_{\mathrm{Id},\mathrm{Id},\Phi}(\boldsymbol{\theta}_i, \boldsymbol{\theta}_j) = \boldsymbol{w}_i^\top \boldsymbol{w}_j + b_i b_j.$

We now derive the kernel corresponding with $\mathrm{Snake}_a$ activation functions up to an offset.

**Kernel 5.** *The kernel $k_{\mathrm{Snake}_a(\cdot)-\frac{1}{2a},\mathrm{Id},\Phi}(\boldsymbol{\theta}_i, \boldsymbol{\theta}_j)$ is equal to*

$$\frac{1}{4a^2}k_{\cos,\mathrm{Id},\Phi}(2a\boldsymbol{\theta}_i, 2a\boldsymbol{\theta}_j) + \boldsymbol{w}_j^\top \boldsymbol{w}_j \Big( \sin(2ab_j)e^{-2a^2\|\boldsymbol{w}_j\|^2} + \sin(2ab_i)e^{-2a^2\|\boldsymbol{w}_i\|^2} \Big)$$

$$- \frac{b_i}{2a}\cos(2ab_j)e^{-2a^2\|\boldsymbol{w}_j\|^2} - \frac{b_j}{2a}\cos(2ab_i)e^{-2a^2\|\boldsymbol{w}_i\|^2} + k_{\mathrm{Id},\mathrm{Id},\Phi}(\boldsymbol{\theta}_i^{(1)}, \boldsymbol{\theta}_j^{(1)}).$$

The kernel corresponding with $\mathrm{Snake}_a$ activations is then stated in terms of Kernel 4 and 5.

**Kernel 6.** *The kernel $k_{\mathrm{Snake}_a,\mathrm{Id},\Phi}(\boldsymbol{\theta}_i, \boldsymbol{\theta}_j)$ is equal to*

$$\frac{1}{2a}\Big( b_i - \frac{1}{2a}\cos(2ab_i)\exp(-2a^2\|\boldsymbol{w}_i\|^2) + b_j - \frac{1}{2a}\cos(2ab_j)\exp(-2a^2\|\boldsymbol{w}_j\|^2) \Big)$$

$$+ k_{\mathrm{Snake}_a(\cdot)-\frac{1}{2a},\mathrm{Id},\Phi}(\boldsymbol{\theta}_i, \boldsymbol{\theta}_j) + \frac{1}{4a^2}.$$

## 3 Properties of squared neural families

### 3.1 Fisher-Neyman factorisation and sufficient statistics

If the base measure $\mu$ is absolutely continuous with respect to some measure $\nu$, and $\frac{d\mu}{d\nu} : \Omega \to [0, \infty)$ is the Radon-Nikodym derivative, then the SNEFY admits a probability density function $p(\cdot \mid \boldsymbol{V}, \boldsymbol{\Theta})$ with respect to $\nu$,

$$p(\boldsymbol{x} \mid \boldsymbol{V}, \boldsymbol{\Theta}) = \underbrace{\frac{d\mu}{d\nu}(\boldsymbol{x})}_{\text{Independent of } \boldsymbol{V}, \boldsymbol{\Theta}} \times \underbrace{\frac{\big\|\boldsymbol{f}\left(\boldsymbol{t}(\boldsymbol{x}); \boldsymbol{V}, \boldsymbol{\Theta}\right)\big\|_2^2}{\mathrm{z}(\boldsymbol{V}, \boldsymbol{\Theta})}}_{\text{Depends on } \boldsymbol{x} \text{ only through } \boldsymbol{t}(\boldsymbol{x})}. \tag{7}$$

The Fisher-Neyman theorem (for example, see Theorem 6.14 of [6]) says that the existence of such a factorisation is equivalent to the fact that $\boldsymbol{t}$ is a sufficient statistic for the parameters $\boldsymbol{V}, \boldsymbol{\Theta}$.

### 3.2 Connections with exponential families

In this section we will use the activation $\sigma(u) = \exp(u/2)$. We note that we can absorb the bias terms $\boldsymbol{b}$ into the $\boldsymbol{V}$ parameters[4] and obtain as a special case the following family of distributions

$$P(d\boldsymbol{x}; \boldsymbol{V}, \boldsymbol{W}) = \frac{1}{\mathrm{Tr}(\boldsymbol{V}^\top \boldsymbol{V} \boldsymbol{K}_{\boldsymbol{\Theta}})} \sum_{i=1}^{n} \sum_{j=1}^{n} \boldsymbol{v}_{\cdot,i}^\top \boldsymbol{v}_{\cdot,j} \exp\left( \frac{1}{2}(\boldsymbol{w}_i + \boldsymbol{w}_j)^\top \boldsymbol{t}(\boldsymbol{x}) \right) \mu(d\boldsymbol{x}), \tag{8}$$

---

[4]Note that $\exp(b_i)$ is independent of $\boldsymbol{x}$ and only appears as a product with $v_i$.

which is a mixture[5] of distributions $P_e\left(\cdot; \frac{1}{2}(\boldsymbol{w}_i + \boldsymbol{w}_j)\right)$ belonging to a classical exponential family $P_e$ [50, 32], given in the canonical form by

$$P_e(d\boldsymbol{x}; \boldsymbol{w}) = \frac{1}{z_e(\boldsymbol{w})} \exp\left(\boldsymbol{w}^\top \boldsymbol{t}(\boldsymbol{x})\right)\mu(d\boldsymbol{x}), \quad z_e(\boldsymbol{w}) = \int_{\mathbb{X}} \exp\left(\boldsymbol{w}^\top \boldsymbol{t}(\boldsymbol{x})\right)\mu(d\boldsymbol{x}). \tag{9}$$

It is helpful to identify the following three further cases:

1. When $n = m = 1$, $v_{11}^2$ cancels in the numerator and denominator and we obtain an exponential family with base measure $\mu$ supported on $\mathbb{X}$, sufficient statistic $\boldsymbol{t}$, canonical parameter $\boldsymbol{w}_1$ and normalising constant $z_e(\boldsymbol{w}_1)$. Every exponential family is thus a SNEFY, but not conversely.

2. When $m > 1$ and $n > 1$, we obtain a type of exponential family mixture model with coefficients $\boldsymbol{V}^\top \boldsymbol{V}$, some of which may be negative. Advantages of allowing negative weights in mixture models in terms of learning rates are discussed in [42]. The rank of $\boldsymbol{V}^\top \boldsymbol{V}$ is at most $\min(m, n)$.

3. When $m > 1$ and $n > 1$ and $\boldsymbol{V}^\top \boldsymbol{V}$ is diagonal (i.e. each column in $\boldsymbol{V}$ is orthogonal), there are at most $n$ non-zero mixture coefficients, all of which are nonnegative. That is, we obtain a standard exponential family mixture model.

The kernel matrix $\boldsymbol{K}_\Theta$ in the normalising constant of (8) is tractable whenever the normalising constant of the corresponding exponential family is itself tractable.

**Proposition 1.** *Denote by* $z_e(\boldsymbol{w})$ *the normalising constant of the exponential family in* (9). *Then*

$$k_{\exp(\cdot/2), \boldsymbol{t}, \mu}(\boldsymbol{w}_i, \boldsymbol{w}_j) = z_e\left(\frac{1}{2}(\boldsymbol{w}_i + \boldsymbol{w}_j)\right). \tag{10}$$

The above kernel is well defined for any collection of $\boldsymbol{w}_i$ which belong to the canonical parameter space of $P_e$, since the canonical parameter space is always convex [50]. This gives us a large number of tractable instances of SNEFY which correspond to exponential family mixture models allowing negative weights – a selection of examples is given in Appendix C. It is interesting that some properties of the exponential families are retained by this generalisation belonging to SNEFYs. For example, the following proposition, the proof of which is given in Appendix A, links the derivatives of the log-normalising constant to the mean and the covariance of the sufficient statistic.

**Proposition 2.** *Let* $\sigma(u) = \exp(u/2)$ *and define the log-normalising constant as* $\Psi = \log z(\boldsymbol{V}, \boldsymbol{\Theta})$.

$$\text{Then} \quad \sum_{i=1}^{n} \frac{\partial \Psi}{\partial \boldsymbol{w}_i} = \mathbb{E}\left[\boldsymbol{t}(\mathsf{x})\right] \quad \text{and} \quad \sum_{i=1}^{n}\sum_{j=1}^{n} \frac{\partial^2 \Psi}{\partial \boldsymbol{w}_i \boldsymbol{w}_j^\top} = \mathbb{E}\left[\boldsymbol{t}(\mathsf{x})\boldsymbol{t}(\mathsf{x})^\top\right] - \mathbb{E}\left[\boldsymbol{t}(\mathsf{x})\right]\mathbb{E}\left[\boldsymbol{t}(\mathsf{x})\right]^\top.$$

### 3.3 Conditional distributions under SNEFY

An attractive property of SNEFY is that, under mild conditions, the family is closed under conditioning.

**Theorem 1.** *Let* $\mathsf{x} = (\mathsf{x}_1, \mathsf{x}_2)$ *be jointly* SNEFY$_{\mathbb{X}_1 \times \mathbb{X}_2, \boldsymbol{t}, \sigma, \mu}$ *with parameters* $\boldsymbol{V}$ *and* $\boldsymbol{\Theta} = ([\boldsymbol{W}_1, \boldsymbol{W}_2], \boldsymbol{b})$. *Assume that* $\mu(d\boldsymbol{x}) = \mu_1(d\boldsymbol{x}_1)\mu_2(d\boldsymbol{x}_2)$ *and* $\boldsymbol{t}(\boldsymbol{x}) = (\boldsymbol{t}_1(\boldsymbol{x}_1), \boldsymbol{t}_2(\boldsymbol{x}_2))$. *Then the conditional distribution of* $\mathsf{x}_1$ *given* $\mathsf{x}_2 = \boldsymbol{x}_2$ *is* SNEFY$_{\mathbb{X}_1, \boldsymbol{t}_1, \sigma, \mu_1}$ *with parameters* $\boldsymbol{V}$ *and* $\boldsymbol{\Theta}_{1|2} \triangleq (\boldsymbol{W}_1, \boldsymbol{W}_2 \boldsymbol{t}_2(\boldsymbol{x}_2) + \boldsymbol{b})$.

The proof, which we detail in Appendix A follows directly by folding the dependence on the conditioning variable $\boldsymbol{x}_2$ into the bias term. We note that conditional density will typically be tractable if the joint density is tractable since they share the same activation function $\sigma$. Thus, whenever SNEFY corresponds to a tractable NNK with a non-zero bias, we can construct highly flexible *conditional density models* using SNEFY by taking $\boldsymbol{t}_2$ itself to be a jointly trained deep neural network. Crucially, $\boldsymbol{t}_2$ may be *completely unconstrained*. We use this observation in the experiments (§ 4).

---

[5]Here we are concerned with the setting where every term in the mixture model belongs to the same family, i.e. an exponential family mixture model but not a mixture of distinct exponential families.

### 3.4 Marginal distributions under SNEFY

Marginal distributions under SNEFY model for a general activation function $\sigma$ need not belong to the same family. In the special case $\sigma = \exp(\cdot/2)$, SNEFY is in fact also closed under marginalisation, which we prove in Appendix D. Even in the general $\sigma$ case, marginal distributions are tractable and admit closed forms whenever the joint SNEFY model and the conditional SNEFY are tractable.

**Theorem 2.** *Let* $\mathbf{x} = (\mathbf{x}_1, \mathbf{x}_2)$ *be jointly* $SNEFY_{\mathbb{X}_1 \times \mathbb{X}_2, \boldsymbol{t}, \sigma, \mu}$ *with parameters* $\boldsymbol{V}$ *and* $\boldsymbol{\Theta} = ([\boldsymbol{W_1}, \boldsymbol{W_2}], \boldsymbol{b})$. *Assume that* $\mu(d\boldsymbol{x}) = \mu_1(d\boldsymbol{x}_1)\mu_2(d\boldsymbol{x}_2)$ *and* $\boldsymbol{t}(\boldsymbol{x}) = (\boldsymbol{t}_1(\boldsymbol{x}_1), \boldsymbol{t}_2(\boldsymbol{x}_2))$. *Then the marginal distribution of* $\mathbf{x}_1$ *is*

$$P_1(d\boldsymbol{x}_1) = \frac{\mathrm{Tr}\left(\boldsymbol{V}^\top \boldsymbol{V} \widetilde{C}_{\boldsymbol{\Theta}}(\boldsymbol{x}_1)\right)}{\mathrm{z}(\boldsymbol{V}, \boldsymbol{\Theta})} \mu_1(d\boldsymbol{x}_1),$$

*where* $\widetilde{C}(\boldsymbol{x}_1)_{ij} = k_{\sigma, \boldsymbol{t}_2, \mu_2}\Big(\big(\boldsymbol{w}_{2i}, \boldsymbol{w}_{1i}^\top \boldsymbol{t}_1(\boldsymbol{x}_1) + b_i\big), \big(\boldsymbol{w}_{2j}, \boldsymbol{w}_{1j}^\top \boldsymbol{t}_1(\boldsymbol{x}_1) + b_j\big)\Big)$.

The proof is given in Appendix A. Due to this tractability of the marginal distributions, it is straightforward to include the likelihood corresponding to incomplete observations (i.e. samples where we are missing some of the components of $\mathbf{x}$) into the density estimation task.

### 3.5 Connections with kernel-based methods for nonnegative functions

SNEFY may be viewed as a neural network variant of the non-parametric kernel models for non-negative functions [25], which are constructed as follows. Let $\boldsymbol{\psi} : \mathbb{X} \to \mathbb{H}$ be a feature mapping to a (possibly infinite dimensional) Hilbert space $\mathbb{H}$. Let $\mathbb{S}(\mathbb{H})$ be the set of all positive semidefinite (PSD) bounded linear operators $\boldsymbol{A} : \mathbb{H} \to \mathbb{H}$. Then

$$h_{\boldsymbol{A}}(\boldsymbol{x}) = \langle \boldsymbol{\psi}(\boldsymbol{x}), \boldsymbol{A}\boldsymbol{\psi}(\boldsymbol{x}) \rangle_{\mathbb{H}} \qquad (11)$$

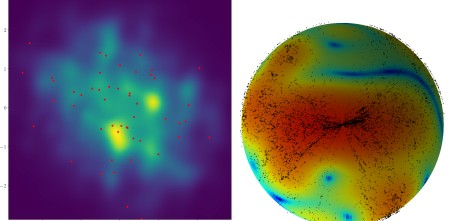

Figure 1: (Left) An instance of an (untrained) $\text{SNEFY}_{\mathbb{R}^2, \mathrm{Id}, \mathrm{Snake}_a, \Phi}$ density with $n = 100$, $m = 1$, $v_{ij} \sim \mathcal{N}(0, 1/n)$, $w_{ij} \sim \mathcal{N}(0, 4)$ and $\boldsymbol{b} = \boldsymbol{0}$. Shown are 50 exact samples found using rejection sampling. Numerical quadrature for this and every example supported on $\mathbb{R}^d$ in § 4 returns a value of 1.00 for the integral over $\mathbb{X}$. (Right) A trained $\text{SNEFY}_{\mathbb{S}^2, \mathrm{Id}, \exp, d\boldsymbol{x}}$ density with $n = m = 30$. Shown is the training and testing dataset [44] also used by [19] for point processes.

gives an elegant model for nonnegative functions parametrised by $\boldsymbol{A} \in \mathbb{S}(\mathbb{H})$, and their application to density modelling with respect to a base measure has also been explored [25, 42]. Note that by assuming boundedness of $\boldsymbol{A}$ and $\boldsymbol{\psi}$, the normalizing constant [25, Proposition 4] is given by $\mathrm{Tr}\left(\boldsymbol{A} \int_{\mathbb{X}} \boldsymbol{\psi}(\boldsymbol{x}) \otimes \boldsymbol{\psi}(\boldsymbol{x})\mu(d\boldsymbol{x})\right)$, which is analogous to our work where the normalising constant is given by $\mathrm{z}(\boldsymbol{V}, \boldsymbol{\Theta}) = \mathrm{Tr}(\boldsymbol{V}^\top \boldsymbol{V} \boldsymbol{K}_{\boldsymbol{\Theta}})$. This can be seen by replacing $\boldsymbol{A}$ with $\boldsymbol{V}^\top \boldsymbol{V}$ and replacing $\int_{\mathbb{X}} \boldsymbol{\psi}(\boldsymbol{x}) \otimes \boldsymbol{\psi}(\boldsymbol{x})\mu(d\boldsymbol{x})$ by $\boldsymbol{K}_{\boldsymbol{\Theta}} = \int_{\mathbb{X}} \sigma(\boldsymbol{W}\boldsymbol{t}(\boldsymbol{x}) + \boldsymbol{b})\sigma(\boldsymbol{W}\boldsymbol{t}(\boldsymbol{x}) + \boldsymbol{b})^\top \mu(d\boldsymbol{x})$.

Despite feature maps being infinite-dimensional, model (11) often reduces to an equivalent representation in finite-dimensions. [25] utilise a representer theorem when (11) is fitted to data using a regularised objective, while [42] more directly assume that the linear operator $\boldsymbol{A}$ is inside the span of the features evaluated at the available data $\{\boldsymbol{x}_\ell\}_{\ell=1}^N$. The resulting model resembles SNEFY where $n$ equals to the number $N$ of datapoints, i.e.

$$h_{\boldsymbol{M}}(\boldsymbol{x}) = [\kappa(\boldsymbol{x}, \boldsymbol{x}_1), \ldots, \ldots, \kappa(\boldsymbol{x}, \boldsymbol{x}_N)]^\top \boldsymbol{M}[\kappa(\boldsymbol{x}, \boldsymbol{x}_1), \ldots, \ldots, \kappa(\boldsymbol{x}, \boldsymbol{x}_N)] \qquad (12)$$

for a PSD matrix $\boldsymbol{M} \in \mathbb{R}^{N \times N}$ and $\kappa(\boldsymbol{x}_i, \boldsymbol{x}_j) = \langle \boldsymbol{\psi}(\boldsymbol{x}_i), \boldsymbol{\psi}(\boldsymbol{x}) \rangle_{\mathbb{H}}$.

However, there are fundamental differences between (12) and SNEFY which we list below. The models can be seen as complementary and they inherit advantages and disdvantages of kernel methods and neural networks common in other settings, respectively.

- **Tractability.** Whereas we identify many tractable instances of SNEFY, the normalising constant of (12) requires computing $\int \kappa(\boldsymbol{x}_i, \boldsymbol{x})\kappa(\boldsymbol{x}, \boldsymbol{x}_j)\mu(d\boldsymbol{x})$ – this is not generally tractable apart from some

limited combinations of $\kappa$ and $\mu$ (e.g. for a Gaussian kernel $\kappa$ and a Gaussian $\mu$). Note that this kernel is evaluated at the datapoints, whereas SNEFY evaluates the kernel at the learned parameters $\boldsymbol{w}_i$. [42] focuses on the specific case where $\kappa$ is a Gaussian kernel, studying properties of the resulting density class which is a mixture of Gaussian densities allowing for negative weights, a model equivalent to SNEFY with the exponential activation function as described in Appendix C. Note that our treatment of SNEFY as a generalisation of the exponential family goes well beyond the Gaussian case, and that tractable instances arise with many other activation functions.

- **Expressivity.** Crucially, the feature map $\boldsymbol{\psi}$ and consequently finite dimensional representation via $\kappa$ in (12) are treated as fixed feature maps and are not themselves learned – instead, expressivity in (12) only comes from fitting $\boldsymbol{M}$ (and potentially lengthscale hyperparameters of $\kappa$), at the expense of a more involved optimiisation over the space of PSD matrices. In contrast, we learn $\boldsymbol{W}$ (analogous to learning $\boldsymbol{\psi}$) and $\boldsymbol{V}$ jointly using neural-network style gradient optimisers. SNEFY is fully compatible with end-to-end and jointly optimised neural network frameworks, a property we leverage heavily in our experiments in § 4.

- **Conditioning.** By explicitly writing parametrisation which includes the biases, we obtain a family closed under conditioning and thus a natural model for conditional densities, whereas it is less clear how to approach conditioning when given a generic feature map $\boldsymbol{\psi}$.

**Other related work**  After submission, we became aware of another related literature, which includes mixture models with potentially negative mixture coefficients via squaring [23] and positive semi-definite probabilistic circuits [43]. We believe a marriage of ideas from SNEFY and probabilistic circuits will lead to future developments in tractable and expressive probability models.

## 4 Experiments

**Implementation**  All experiments are conducted on a Dual Xeon 14-core E5-2690 with 30GB of reserved RAM and a single NVidia Tesla P100 GPU. Full experimental details are given in Appendix E. We build our implementation on top of `normflows` [45], a PyTorch package for normalising flows. SNEFYs are built as a `BaseDistribution`, which are base distributions inside a `NormalizingFlow` with greater than or equal to zero layers. We train all models via maximum likelihood estimation (MLE) i.e. minimising forward KL divergence.

**2D synthetic unconditional density estimation**  We consider the 2 dimensional problems also benchmarked in [46]. We compare the test performance, computation time and parameter count of non-volume preserving flows (NVPs) [7] with four types of base distribution: SNEFY, resampled [46] (`Resampled`), diagonal Gaussian (`Gauss`) and Gaussian mixture model (`GMM`). We consider flow depths of 0, 1, 2, 4, 8 and 32 where a flow depth of 0 corresponds with the base distribution only. We use $\sigma = \cos$, $\boldsymbol{t} = \mathrm{Id}$, $\mathbb{X} = \mathbb{R}^d$ and a Gaussian mixture model base density. We set $m = 1$ and $n = 50$. Full architectures and further experimental details are described in Appendix E.1.

Results are shown in Table 2 for the 0 and 16 layer cases, and further tables for $1, 2, 4$ and $8$ layers are given in Appendix E.1. Our observations are that all base distributions are able to achieve good performance, provided they are appended with a normalising flow of appropriate depth. SNEFY is able to achieve good performance with depth 0 on all three datasets, as are `Resampled` and `GMM` for the Moons dataset. The parameter count for well-performing SNEFY models is very low, but the computation time can be relatively high. However, SNEFY is the only model which consistently achieves the highest performance within one standard deviation across all normalising flow depths.

**Data on the sphere**  We compare the three cases mentioned in § 3.2 using Kernel 3, i.e. mixtures of the von Mises Fisher (VMF) distribution, as shown in Figure 1. Over 50 runs, the unconstrained $\boldsymbol{V}$, diagonal $\boldsymbol{V}$, and $n = m = 1$ cases respectively obtain test negative log likelihoods of $1.38 \pm 9.64 \times 10^{-3}$, $1.46 \pm 0.016$ and $2.34 \pm 0.26$ each in $111.18 \pm 2.58$, $109.12 \pm 0.80$ and $61.88 \pm 0.33$ seconds (average $\pm$ standard deviation). In this setting, allowing for a fully flexible $\boldsymbol{V}$, going beyond the classical mixture model, shows clear benefits in performance. Results are summarised in Table 3. Full details are given in Appendix E.2.

**Conditional density estimation on astronomy data**  Predicting plausible values for the velocity of distant astronomical objects (such as galaxies or quasars) without measuring their full spectra, but by

| | SNEFY 0 | Resampled 0 | Gauss 0 | GMM 0 | SNEFY 16 | Resampled 16 | Gauss 16 | GMM 16 |
|---|---|---|---|---|---|---|---|---|
| Moons | $-1.59 \pm 0.02$ | $-1.60 \pm 0.02$ | $-3.29 \pm 0.02$ | $-1.59 \pm 0.04$ | $-1.57 \pm 0.03$ | $-1.59 \pm 0.02$ | $-1.68 \pm 0.24$ | $-1.57 \pm 0.03$ |
| | 241 | 66817 | 4 | 50 | 19281 | 85857 | 19044 | 19090 |
| | $1200.84 \pm 109.77$ | $541.11 \pm 12.20$ | $606.35 \pm 20.29$ | $629.11 \pm 14.57$ | $2844.09 \pm 254.02$ | $1867.26 \pm 167.75$ | $2226.37 \pm 174.80$ | $2276.78 \pm 211.15$ |
| Circles | $-1.92 \pm 0.03$ | $-2.42 \pm 0.22$ | $-3.55 \pm 0.01$ | $-2.07 \pm 0.07$ | $-1.92 \pm 0.04$ | $-1.96 \pm 0.04$ | $-2.22 \pm 0.30$ | $-1.93 \pm 0.03$ |
| | 241 | 66817 | 4 | 50 | 19281 | 85857 | 19044 | 19090 |
| | $643.80 \pm 150.84$ | $166.35 \pm 20.59$ | $52.63 \pm 3.96$ | $73.52 \pm 5.47$ | $2272.15 \pm 281.76$ | $1533.30 \pm 158.90$ | $1660.77 \pm 169.02$ | $1673.54 \pm 166.84$ |
| Rings | $-2.41 \pm 0.05$ | $-2.62 \pm 0.01$ | $-3.26 \pm 0.01$ | $-2.67 \pm 0.03$ | $-2.34 \pm 0.10$ | $-2.36 \pm 0.14$ | $-2.51 \pm 0.22$ | $-2.31 \pm 0.04$ |
| | 241 | 66817 | 4 | 50 | 19281 | 85857 | 19044 | 19090 |
| | $997.73 \pm 107.80$ | $409.77 \pm 19.37$ | $416.23 \pm 18.11$ | $433.82 \pm 16.06$ | $2654.22 \pm 269.14$ | $1774.06 \pm 182.25$ | $2045.65 \pm 170.80$ | $2070.60 \pm 177.06$ |

Table 2: The first quantity is the average $\pm$ sample standard deviation over 20 runs of test loglikelihood. The second quantity is the parameter count. The third quantity is the average $\pm$ sample standard deviation over 20 runs of the computation time (seconds). The number in each column header is the number of non-volume preserving flow layers appended after the base distribution. Here there are 0 or 16 NVP layers. More tables showing intermediate layer results are given in Appendix E.1.

| Method | Average Test NLL $\downarrow$ | Compute time (s) | Parameter count |
|---|---|---|---|
| SNEFY $n = 32$ | $-2.195 \pm 0.024$ | $495.720 \pm 22.561$ | 179748 |
| SNEFY $n = 16$ | $-2.172 \pm 0.034$ | $404.291 \pm 57.605$ | 46356 |
| SNEFY $n = 8$ | $-2.108 \pm 0.089$ | $390.870 \pm 16.028$ | 12300 |
| CNF $L = 4$ | $-2.156 \pm 0.018$ | $202.1290 \pm 10.380$ | 1413 |
| CNF $L = 2$ | $-2.163 \pm 0.024$ | $155.090 \pm 15.809$ | 1155 |
| CNF $L = 1$ | $-2.171 \pm 0.012$ | $122.304 \pm 1.194$ | 1026 |
| CKDE | $-2.148$ | 391.867 (Not GPU-accelerated) | Train set size = $74309 \times 6$ |

Table 3: Performance comparison of methods on astronomy dataset. Excluding CKDE which is deterministic, an average is taken over 50 random initialisations with $\pm$ indicating standard deviation. SNEFY shows a statistically significant average increase in performance over CNF (using a two-sample t-test), and over CKDE (using a one-sample t-test). Full experimental details in Appendix E.3.

measuring shifted spectra through broadband filters, is known as photometric redshift. Photometric redshift is an important problem in modern astronomy, as large surveys have increased the amount of available data that do not directly measure spectra. We estimate distributions for cosmological redshift $x_1 \in \mathbb{R}$ conditional on features $x_2 \in \mathbb{R}^5$ using a public dataset [2]. The features $x_2$ are the magnitude a broadband filter (red) and set of four pairwise distances between broadband colour filters (ultraviolet-green, green-red, red-infrared1, infrared1-infrared2). We benchmark our own SNEFY against a conditional kernel density estimator (CKDE) and a conditional normalising flow (CNF) [54]. We use a publically available implementation [10] of CNF. Our SNEFY here is not appended with any normalising flow layers. We use $\sigma = \text{Snake}_a$, $t = \text{Id}$, $\mathbb{X} = \mathbb{R}$ and a Gaussian base density. We use a deep conditional feature extractor with layer widths $[2^5 n, 2^4 n, 2^3 n, 2^2 n, 2n]$, and then set $m = n/2$. Full details are given in Appendix E.3.

While CNFs have shown great promise in modelling high dimensional image data, we expect that they are not as well-suited to nontrivial tabular data with a small to medium number of dimensions. This is because each layer is required to be an invertible transformation in the variable being modelled, so the input and output sizes must be the same, thereby significantly limiting the parameter count in each layer. We could of course increase the number of layers in the CNF model to achieve higher parameter counts, however this results in a model that is difficult to train. In our experiments, we found that increasing the depth of the CNF decreased its performance. On the other hand, SNEFY may utilise any number of parameters, as the conditioning network $t_2$ is completely unrestricted.

**Density estimation on astronomy data using partial (marginal) observations**  We perform joint probability density estimation using the redshift data. The original training dataset is a matrix in $\mathbb{R}^{74309 \times 6}$. We partition this matrix into batches of size 256 and randomly set each column of each batch to NaN with probability $q$ (i.e. the corresponding dimension is missing from observations). We then train a SNEFY model that utilises partial observations by maximising the marginal likelihood according to Theorem 2. We leave the original $74557 \times 6$ testing dataset untouched, and measure the test NLL after training. We plot the NLL as a function of $1 - q$, as shown in Figure 2. We also compare the performance of the SNEFY that uses partial observations with the performance of SNEFY that simply throws away incomplete observations, as well as the same normalising flow baselines that throw away incomplete observations that were used in the conditional density estimation setting. There do exist normalising flow approaches that jointly optimises conditional distributions for missing data and

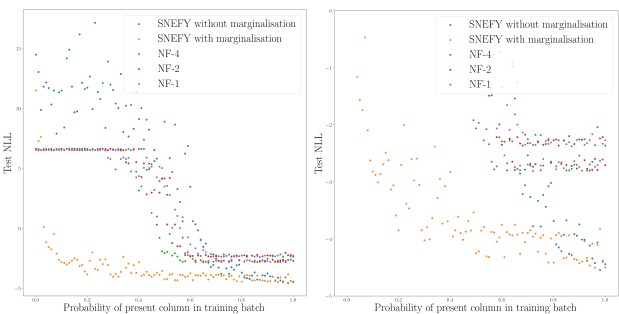

Figure 2: Density estimation under partial observations. The right plot is a zoomed in version of the left plot. NF-1, NF-2 and NF-4 are respectively normalising flows of depth 1, 2 and 4. Normalising flow models and `SNEFY` without marginalisation discard incomplete observations, whereas `SNEFY` use Theorem 2 to include partial observations via maximum marginal likelihood. Marginal likelihoods allow for an improved NLL.

model parameters [3] or samples from missing data and optimises for model parameters [41], however these do not allow for maximum likelihood estimation. Our observations are twofold: including partial observations improves performance, and adding more complete observations improves performance.

## 5 Discussion and conclusion

We constructed a new class of probability models – `SNEFY` – by normalising the squared norm of a neural network with respect to a base measure. `SNEFY` possesses a number of convenient properties: tractable exact normalising constants in many instances we identified, closure under conditioning, tractable marginalisation, and intriguing connections with existing models. `SNEFY` shows promise empirically, outperforming competing (conditional) density estimation methods in experiments.

**Sampling versus density estimation**    We focus here on the problem of density estimation, for which `SNEFY` is well-suited. While it is sometimes possible to obtain exact `SNEFY` samples using rejection sampling, sampling is more computationally expensive than in other models such as normalising flows. Future work will focus on sampling, as has been done with related models [26], where the special case of Gaussian $\sigma$ and hyperrectangular support $\mathbb{X}$ is considered. In [26], $r$ approximate samples with $\mathcal{O}\left(r \log_2 |\mathbb{X}| + rd \log_2 \frac{2}{\rho}\right)$ evaluations of the normalising constant are obtained. Here $\rho$ is an approximation tolerance parameter. Note that this complexity significantly improves upon naive rejection sampling, which typically scales exponentially in dimension $d$. In the unconditional setting, `SNEFY` is constructed as only a 2-layer network, thereby limiting expressivity. However, in the conditional density estimation setting, we may use any number of layers and any architecture for the conditioning network $t_2$. Finally, `SNEFY` inherits all the usual limitations and advantages over mirroring kernel-based approaches, as discussed in § 3.5.

**Future work**    We see a number of further promising future research directions. First, as we detail in Appendix F, choosing $\sigma(\cdot) = \exp(i\cdot)$ and identity sufficient statistics results in a kernel $k_{\exp(i\cdot),\mathrm{Id},\mu}$ which is the Fourier transform of a nonnegative measure. By Bochner's theorem, the kernel is guaranteed to be (real or complex-valued) shift-invariant. The kernel matrix is Hermitian PSD (so that the normalising constant is positive and nonnegative), and we may also allow (but do not require) the readout parameters $V$ to be complex. We note that the same result would be obtained if one used a mixture of real-valued cos and sin activations with shared parameters (see Remark 3). Second, an alternative deep model to our deep conditional feature extractor might be to use a `SNEFY` model as a base measure $\mu$ for another `SNEFY` model; this might be repeated $L$ times. This leads to $\mathcal{O}(n^{2L})$ integration terms for the normalising constant. The individual terms are tractable in certain cases, for example when $\sigma$ is exponential or trigonometric. Third, when modelling discrete distributions with trigonometric activations, the NNK can be expressed in terms of convergent Fourier series (see Appendix G) Finally, our integration technique can be applied to other settings. For example, we may build a Poisson point process intensity function using a squared neural network and compute the intensity function in closed-form, offering a model that scales quadratically in the number of neurons $\mathcal{O}(n^2)$ instead of comparable models which scale cubically in the number of datapoints $\mathcal{O}(N^3)$ [9, 51].

## Acknowledgments and Disclosure of Funding

Russell and Cheng Soon would like to acknowledge the support of the Machine Learning and Artificial Intelligence Future Science Platform, CSIRO. The authors would like to thank Jia Liu for early discussions about the idea.

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

# A Proofs

**Identity 1.** *The integral* (2) *admits a representation of the form*

$$z(\boldsymbol{V}, \boldsymbol{\Theta}) = \operatorname{Tr}\left(\boldsymbol{V}^\top \boldsymbol{V} \boldsymbol{K_\Theta}\right) \tag{6}$$

*where $k_{\sigma, \boldsymbol{t}, \mu}$ is as defined in* (5), *and $\boldsymbol{K_\Theta}$ is the PSD matrix whose $ij$th entry is $k_{\sigma, \boldsymbol{t}, \mu}(\boldsymbol{\theta}_i, \boldsymbol{\theta}_j)$.*

*Proof.* Let $\widetilde{\boldsymbol{K_\Theta}}(\boldsymbol{x})$ be the PSD matrix whose $ij$th entry is $\sigma(\boldsymbol{w}_i^\top \boldsymbol{t}(x) + b_i)\sigma(\boldsymbol{w}_j^\top \boldsymbol{t}(x) + b_j)^\top$. The squared norm of the neural network evaluation is given by $\operatorname{Tr}\left(\boldsymbol{V}^\top \boldsymbol{V} \widetilde{\boldsymbol{K_\Theta}}(\boldsymbol{x})\right)$, since

$$
\begin{aligned}
\left\|\boldsymbol{V}\sigma(\boldsymbol{W}\boldsymbol{t}(\boldsymbol{x}) + \boldsymbol{b})\right\|_2^2 &= \sum_{i=1}^m \sum_{j_1=1}^n \sum_{j_2=1}^n v_{ij_1} v_{ij_2} \sigma\left(\boldsymbol{w}_{j_1}^\top \boldsymbol{t}(x) + b_{j_1}\right)\sigma\left(\boldsymbol{w}_{j_2}^\top \boldsymbol{t}(x) + b_{j_2}\right) \\
&= \sum_{j_1=1}^n \sum_{j_2=1}^n \boldsymbol{v}_{\cdot,j_1}^\top \boldsymbol{v}_{\cdot,j_2} \sigma\left(\boldsymbol{w}_{j_1}^\top \boldsymbol{t}(x) + b_{j_1}\right)\sigma\left(\boldsymbol{w}_{j_2}^\top \boldsymbol{t}(x) + b_{j_2}\right) \\
&= \langle \boldsymbol{V}^\top \boldsymbol{V}, \widetilde{\boldsymbol{K_\Theta}}(\boldsymbol{x})\rangle_F \\
&= \operatorname{Tr}\left(\boldsymbol{V}^\top \boldsymbol{V} \widetilde{\boldsymbol{K_\Theta}}(\boldsymbol{x})\right),
\end{aligned}
$$

where $\boldsymbol{v}_{\cdot,j_1}$ denotes the $j_1$th column of $\boldsymbol{V}$ and $\langle \cdot, \cdot \rangle_F$ denotes the Frobenius inner product. Therefore using the definition (2) directly and the linearity of the Frobenius inner product, the normalising constant is

$$
\begin{aligned}
z(\boldsymbol{V}, \boldsymbol{\Theta}) &= \int_{\mathbb{X}} \left\|\boldsymbol{V}\sigma(\boldsymbol{W}\boldsymbol{t}(\boldsymbol{x}) + \boldsymbol{b})\right\|_2^2 \mu(d\boldsymbol{x}) \\
&= \int_{\mathbb{X}} \operatorname{Tr}\left(\boldsymbol{V}^\top \boldsymbol{V} \widetilde{\boldsymbol{K_\Theta}}(\boldsymbol{x})\right)\mu(d\boldsymbol{x}) \\
&= \operatorname{Tr}\left(\boldsymbol{V}^\top \boldsymbol{V} \int_{\mathbb{X}} \widetilde{\boldsymbol{K_\Theta}}(\boldsymbol{x})\mu(d\boldsymbol{x})\right) \\
&= \operatorname{Tr}\left(\boldsymbol{V}^\top \boldsymbol{V} \boldsymbol{K_\Theta}\right). \tag{13}
\end{aligned}
$$

$\square$

**Proposition 2.** *Let $\sigma(u) = \exp(u/2)$ and define the log-normalising constant as $\Psi = \log z(\boldsymbol{V}, \boldsymbol{\Theta})$.*

*Then* $\quad \displaystyle\sum_{i=1}^n \frac{\partial \Psi}{\partial \boldsymbol{w}_i} = \mathbb{E}\left[\boldsymbol{t}(\mathsf{x})\right] \quad$ *and* $\quad \displaystyle\sum_{i=1}^n \sum_{j=1}^n \frac{\partial^2 \Psi}{\partial \boldsymbol{w}_i \boldsymbol{w}_j^\top} = \mathbb{E}\left[\boldsymbol{t}(\mathsf{x})\,\boldsymbol{t}(\mathsf{x})^\top\right] - \mathbb{E}\left[\boldsymbol{t}(\mathsf{x})\right]\mathbb{E}\left[\boldsymbol{t}(\mathsf{x})\right]^\top.$

*Proof.* The result follows by noticing that the logarithmic derivative property holds, $\sum_{i=1}^n \frac{\partial \Psi}{\partial \boldsymbol{w}_i} = \frac{1}{z}\sum_{i=1}^n \frac{\partial z}{\partial \boldsymbol{w}_i}$, and that by writing

$$
z = \sum_{i,j} \boldsymbol{v}_{\cdot,i}^\top \boldsymbol{v}_{\cdot,j} \int \exp\left(\frac{1}{2}\boldsymbol{w}_i^\top \boldsymbol{t}(\boldsymbol{x})\right) \exp\left(\frac{1}{2}\boldsymbol{w}_j^\top \boldsymbol{t}(\boldsymbol{x})\right) \mu(d\boldsymbol{x}),
$$

we obtain

$$
\begin{aligned}
\frac{\partial z}{\partial \boldsymbol{w}_i} &= \|\boldsymbol{v}_{\cdot,i}\|^2 \int \boldsymbol{t}(\boldsymbol{x}) \exp\left(\boldsymbol{w}_i^\top \boldsymbol{t}(\boldsymbol{x})\right) \mu(d\boldsymbol{x}) \\
&\quad + \boldsymbol{v}_{\cdot,i}^\top \sum_{j\neq i} \boldsymbol{v}_{\cdot,j} \int \boldsymbol{t}(\boldsymbol{x}) \exp\left(\frac{1}{2}\boldsymbol{w}_i^\top \boldsymbol{t}(\boldsymbol{x})\right) \exp\left(\frac{1}{2}\boldsymbol{w}_j^\top \boldsymbol{t}(\boldsymbol{x})\right) \mu(d\boldsymbol{x}) \\
&= \boldsymbol{v}_{\cdot,i}^\top \sum_j \boldsymbol{v}_{\cdot,j} \int \boldsymbol{t}(\boldsymbol{x}) \exp\left(\frac{1}{2}\boldsymbol{w}_i^\top \boldsymbol{t}(\boldsymbol{x})\right) \exp\left(\frac{1}{2}\boldsymbol{w}_j^\top \boldsymbol{t}(\boldsymbol{x})\right) \mu(d\boldsymbol{x}).
\end{aligned}
$$

Now summing across $i$ gives $\sum_{i=1}^n \frac{\partial \Psi}{\partial \boldsymbol{w}_i} = \int \boldsymbol{t}(\boldsymbol{x})P(d\boldsymbol{x})$ as required.

To obtain the second result, we apply the product rule to find

$$\sum_{i,j} \frac{\partial^2 \Psi}{\partial \boldsymbol{w}_i \boldsymbol{w}_j^\top} = \frac{1}{\mathrm{z}} \sum_{i,j} \frac{\partial^2 \mathrm{z}}{\partial \boldsymbol{w}_i \boldsymbol{w}_j^\top} - \left( \frac{1}{\mathrm{z}} \sum_i \frac{\partial \mathrm{z}}{\partial \boldsymbol{w}_i} \right) \left( \frac{1}{\mathrm{z}} \sum_j \frac{\partial \mathrm{z}}{\partial \boldsymbol{w}_j^\top} \right)$$

and note that

$$\frac{\partial^2 \mathrm{z}}{\partial \boldsymbol{w}_i \boldsymbol{w}_j^\top} = \begin{cases} \frac{1}{2} \boldsymbol{v}_{.,i}^\top \boldsymbol{v}_{.,j} \int \boldsymbol{t}(\boldsymbol{x}) \boldsymbol{t}(\boldsymbol{x})^\top \exp \left( \frac{1}{2} \boldsymbol{w}_i^\top \boldsymbol{t}(\boldsymbol{x}) \right) \exp \left( \frac{1}{2} \boldsymbol{w}_j^\top \boldsymbol{t}(\boldsymbol{x}) \right) \mu \left( d\boldsymbol{x} \right), & \text{if } i \neq j, \\ \|\boldsymbol{v}_{.,i}\|^2 \int \boldsymbol{t}(\boldsymbol{x}) \boldsymbol{t}(\boldsymbol{x})^\top \exp \left( \boldsymbol{w}_i^\top \boldsymbol{t}(\boldsymbol{x}) \right) \mu \left( d\boldsymbol{x} \right) & \\ \quad + \frac{1}{2} \boldsymbol{v}_{.,i}^\top \sum_{r \neq i} \boldsymbol{v}_{.,j} \int \boldsymbol{t}(\boldsymbol{x}) \boldsymbol{t}(\boldsymbol{x})^\top \exp \left( \frac{1}{2} \boldsymbol{w}_i^\top \boldsymbol{t}(\boldsymbol{x}) \right) \exp \left( \frac{1}{2} \boldsymbol{w}_r^\top \boldsymbol{t}(\boldsymbol{x}) \right) \mu \left( d\boldsymbol{x} \right), & \text{if } i = j. \end{cases}$$

Thus,

$$\begin{aligned}
\frac{1}{\mathrm{z}} \sum_{i,j} \frac{\partial^2 \mathrm{z}}{\partial \boldsymbol{w}_i \boldsymbol{w}_j^\top} &= \frac{1}{\mathrm{z}} \sum_{i=1}^n \left( \frac{\partial^2 \mathrm{z}}{\partial \boldsymbol{w}_i \boldsymbol{w}_i^\top} + \sum_{j \neq i} \frac{\partial^2 \mathrm{z}}{\partial \boldsymbol{w}_i \boldsymbol{w}_j^\top} \right) \\
&= \frac{1}{\mathrm{z}} \sum_{i,j} \boldsymbol{v}_i^\top \boldsymbol{v}_j \int \boldsymbol{t}(\boldsymbol{x}) \boldsymbol{t}(\boldsymbol{x})^\top \exp \left( \frac{1}{2} \boldsymbol{w}_i^\top \boldsymbol{t}(\boldsymbol{x}) \right) \exp \left( \frac{1}{2} \boldsymbol{w}_j^\top \boldsymbol{t}(\boldsymbol{x}) \right) \mu \left( d\boldsymbol{x} \right) \\
&= \int \boldsymbol{t}(\boldsymbol{x}) \boldsymbol{t}(\boldsymbol{x})^\top P(d\boldsymbol{x}),
\end{aligned}$$

as required.

$\square$

**Remark 2.** *Given the above relationship between the log-normalising constant and the expectation of the sufficient statistic, we can also ask whether the maximum likelihood estimation of the mean parameters $\mathbb{E}\left[ \boldsymbol{t}\left( \mathsf{x} \right) \right]$ proceeds in the same way as in the exponential family case. The answer is positive but with two caveats. First, the log-likelihood need not be concave in $\boldsymbol{W}$, and may have many local optima or stationary points. Second, unlike the exponential family distribution, the SNEFY distribution is not determined by its mean parameters, so the MLE estimation of the mean parameters may not constitute a meaningful task in SNEFY modelling (unless we are in the case where precisely the expectation of $\boldsymbol{t}(\boldsymbol{x})$ under the SNEFY model is of interest).*

**Corollary 1.** *Given a dataset $\{\mathsf{x}_\ell\}_{\ell=1}^N$, and a SNEFY model with $\sigma(u) = \exp(\frac{1}{2}u)$, assume that all rows of the maximum likelihood estimator of $\boldsymbol{W}$ are in the interior of the natural parameter space of the corresponding exponential family. Denote the mean parameter as $\boldsymbol{m} = \mathbb{E}\left[ \boldsymbol{t}\left( \mathsf{x} \right) \right]$. Then the maximum likelihood estimate of $\boldsymbol{m}$ is $\hat{\boldsymbol{m}} = \frac{1}{N} \sum_{\ell=1}^N \boldsymbol{t}(\boldsymbol{x}_\ell)$.*

*Proof.* Since MLE is achieved at a stationary point of the log-likelihood, the proof follows by writing the log likelihood as

$$\sum_{\ell=1}^N \log p(\boldsymbol{x}_\ell; \boldsymbol{V}, \boldsymbol{\Theta}) = \text{const} + \sum_{\ell=1}^N \log \|\boldsymbol{f}(\boldsymbol{t}(\boldsymbol{x}_\ell); \boldsymbol{V}, \boldsymbol{\Theta})\|_2^2 - N\Psi$$

and concluding that at the MLE $\{\boldsymbol{w}_i^*\}_{i=1}^n$ for $\boldsymbol{W}$, we must have

$$\sum_{\ell=1}^N \frac{\partial \log \|\boldsymbol{f}(\boldsymbol{t}(\boldsymbol{x}_\ell); \boldsymbol{V}, \boldsymbol{\Theta})\|_2^2}{\partial \boldsymbol{w}_i^*} = N \frac{\partial \Psi}{\partial \boldsymbol{w}_i^*}, \quad i = 1, \ldots, n. \tag{14}$$

But

$$\sum_{i=1}^n \frac{\partial \log \|\boldsymbol{f}(\boldsymbol{t}(\boldsymbol{x}_\ell); \boldsymbol{V}, \boldsymbol{\Theta})\|_2^2}{\partial \boldsymbol{w}_i} = \boldsymbol{t}(\boldsymbol{x}_\ell),$$

so the result follows by summing (14) over $i$ and dividing by $N$. Note that maximum likelihood estimates are invariant to transformations, even if the transformation is not bijective. So if $\{\boldsymbol{w}_i^*\}_{i=1}^n$ is an MLE, we may construct a mapping from $\{\boldsymbol{w}_i^*\}_{i=1}^n$ to a corresponding MLE $\hat{\boldsymbol{m}}$ for the mean parameter. $\square$

**Theorem 1.** *Let* $\mathbf{x} = (\mathbf{x}_1, \mathbf{x}_2)$ *be jointly* $\mathtt{SNEFY}_{\mathbb{X}_1 \times \mathbb{X}_2, t, \sigma, \mu}$ *with parameters* $\boldsymbol{V}$ *and* $\boldsymbol{\Theta} = ([\boldsymbol{W_1}, \boldsymbol{W_2}], \boldsymbol{b})$. *Assume that* $\mu(d\boldsymbol{x}) = \mu_1(d\boldsymbol{x}_1)\mu_2(d\boldsymbol{x}_2)$ *and* $\boldsymbol{t}(\boldsymbol{x}) = (\boldsymbol{t}_1(\boldsymbol{x}_1), \boldsymbol{t}_2(\boldsymbol{x}_2))$. *Then the conditional distribution of* $\mathbf{x}_1$ *given* $\mathbf{x}_2 = \boldsymbol{x}_2$ *is* $\mathtt{SNEFY}_{\mathbb{X}_1, t_1, \sigma, \mu_1}$ *with parameters* $\boldsymbol{V}$ *and* $\boldsymbol{\Theta}_{1|2} \triangleq (\boldsymbol{W_1}, \boldsymbol{W_2}\boldsymbol{t}_2(\boldsymbol{x}_2) + \boldsymbol{b})$.

*Proof.* The joint distribution of $\mathbf{x}$ satisfies

$$P(d\boldsymbol{x}; \boldsymbol{V}, \boldsymbol{\Theta}) \propto \left\| \boldsymbol{V}\sigma\big(\boldsymbol{W_1}\boldsymbol{t}_1(\boldsymbol{x}_1) + \boldsymbol{W_2}\boldsymbol{t}_2(\boldsymbol{x}_2) + \boldsymbol{b}\big) \right\|_2^2 \mu(d\boldsymbol{x}).$$

Therefore, the distribution of $\mathbf{x}_1$ conditionally on $\mathbf{x}_2 = \boldsymbol{x}_2$, which is obtained by dividing the joint distribution by the marginal distribution of $\mathbf{x}_2$ (which is independent of $\boldsymbol{x}_1$), satisfies

$$P_1\Big(d\boldsymbol{x}_1 \mid \boldsymbol{x}_2; \boldsymbol{V}, \big(\boldsymbol{W_1}, \boldsymbol{W_2}\boldsymbol{t}_2(\boldsymbol{x}_2) + \boldsymbol{b}\big)\Big) \propto \Big(\boldsymbol{V}\sigma\big(\boldsymbol{W_1}\boldsymbol{t}_1(\boldsymbol{x}_1) + \boldsymbol{W_2}\boldsymbol{t}_2(\boldsymbol{x}_2) + \boldsymbol{b}\big)\Big)^2 \mu_1(d\boldsymbol{x}_1).$$

That is, the term $\boldsymbol{W_2}\boldsymbol{t}_2(\boldsymbol{x}_2) + \boldsymbol{b}$ is viewed as a constant bias term when the expression on the right hand side is an unnormalised measure with respect to the variable $\boldsymbol{x}_1$. $\qquad\square$

**Theorem 2.** *Let* $\mathbf{x} = (\mathbf{x}_1, \mathbf{x}_2)$ *be jointly* $\mathtt{SNEFY}_{\mathbb{X}_1 \times \mathbb{X}_2, t, \sigma, \mu}$ *with parameters* $\boldsymbol{V}$ *and* $\boldsymbol{\Theta} = ([\boldsymbol{W_1}, \boldsymbol{W_2}], \boldsymbol{b})$. *Assume that* $\mu(d\boldsymbol{x}) = \mu_1(d\boldsymbol{x}_1)\mu_2(d\boldsymbol{x}_2)$ *and* $\boldsymbol{t}(\boldsymbol{x}) = (\boldsymbol{t}_1(\boldsymbol{x}_1), \boldsymbol{t}_2(\boldsymbol{x}_2))$. *Then the marginal distribution of* $\mathbf{x}_1$ *is*

$$P_1(d\boldsymbol{x}_1) = \frac{\mathrm{Tr}\left(\boldsymbol{V}^\top \boldsymbol{V}\widetilde{C}_{\boldsymbol{\Theta}}(\boldsymbol{x}_1)\right)}{\mathrm{z}(\boldsymbol{V}, \boldsymbol{\Theta})}\mu_1(d\boldsymbol{x}_1),$$

*where* $\widetilde{C}(\boldsymbol{x}_1)_{ij} = k_{\sigma, t_2, \mu_2}\Big(\big(\boldsymbol{w}_{2i}, \boldsymbol{w}_{1i}^\top\boldsymbol{t}_1(\boldsymbol{x}_1) + b_i\big), \big(\boldsymbol{w}_{2j}, \boldsymbol{w}_{1j}^\top\boldsymbol{t}_1(\boldsymbol{x}_1) + b_j\big)\Big)$.

*Proof.* The marginal distribution of the random variable $\mathbf{x}_1$ is obtained by marginalising out the joint distribution with respect to $\boldsymbol{x}_2$,

$$P_1(d\boldsymbol{x}_1) = \frac{1}{\mathrm{z}(\boldsymbol{V}, \boldsymbol{\Theta})}\underbrace{\left(\int_{\mathbb{X}_2}\left\|\boldsymbol{V}\sigma\big(\boldsymbol{W_1}\boldsymbol{t}_1(\boldsymbol{x}_1) + \boldsymbol{W_2}\boldsymbol{t}_2(\boldsymbol{x}_2) + \boldsymbol{b}\big)\right\|_2^2\mu_2(d\boldsymbol{x}_2)\right)}_{\triangleq \mathrm{z}_2}\mu_1(d\boldsymbol{x}_1).$$

The integral $\mathrm{z}_2$ takes a similar form to $\mathrm{z}(\boldsymbol{V}, \boldsymbol{\Theta})$,

$$\mathrm{z}_2 = \mathrm{Tr}\left(\boldsymbol{V}^\top \boldsymbol{V}\widetilde{C}_{\boldsymbol{\Theta}}(\boldsymbol{x}_1)\right), \quad \text{where}$$

$$\widetilde{C}_{ij}(\boldsymbol{x}_1) = \int_{\mathbb{X}_2}\sigma\big(\boldsymbol{w}_{1i}^\top\boldsymbol{t}_1(\boldsymbol{x}_1) + \boldsymbol{w}_{2i}^\top\boldsymbol{t}_2(\boldsymbol{x}_2) + b_i\big)\sigma\big(\boldsymbol{w}_{1j}^\top\boldsymbol{t}_1(\boldsymbol{x}_1) + \boldsymbol{w}_{2j}^\top\boldsymbol{t}_2(\boldsymbol{x}_2) + b_j\big)\mu_2(d\boldsymbol{x}_2)$$

$$= k_{\sigma, t_2, \mu_2}\Big(\big(\boldsymbol{w}_{2i}, \boldsymbol{w}_{1i}^\top\boldsymbol{t}_1(\boldsymbol{x}_1) + b_i\big), \big(\boldsymbol{w}_{2j}, \boldsymbol{w}_{1j}^\top\boldsymbol{t}_1(\boldsymbol{x}_1) + b_j\big)\Big).$$

$\qquad\square$

## B  Derivation of neural network kernels

**Kernel 1.** $k_{\sigma, \mathrm{Id}, \Phi_{C,m}}(\boldsymbol{\theta}_i, \boldsymbol{\theta}_j) = k_{\sigma, \mathrm{Id}, \Phi}(\mathcal{T}\boldsymbol{\theta}_i, \mathcal{T}\boldsymbol{\theta}_j)$, *where* $\mathcal{T}\boldsymbol{\Theta} = (\boldsymbol{W}\boldsymbol{A}, \boldsymbol{b} + \boldsymbol{W}\boldsymbol{m})$, $\mathcal{T}\boldsymbol{\theta}_i = (\boldsymbol{w}_i^\top\boldsymbol{A}, b_i + \boldsymbol{w}_i^\top\boldsymbol{m})$ *and* $\boldsymbol{A}$ *is a matrix factor such that covariance* $\boldsymbol{C} = \boldsymbol{A}\boldsymbol{A}^\top$.

*Proof.* The NNK may be expressed as an expectation with respect to a Gaussian random variable $\mathbf{x}$ with mean $\boldsymbol{m}$ and covariance matrix $\boldsymbol{C}$. It holds that $\mathbf{x} \stackrel{d}{=} \boldsymbol{A}\mathbf{z} + \boldsymbol{m}$, where $\mathbf{z}$ is a zero-mean independent standard Gaussian random vector, so the kernel may be expressed in terms of an expectation over $\mathbf{z}$ instead. More concretely,

$$\begin{aligned}
k_{\sigma, \mathrm{Id}, \Phi_{C,m}}(\boldsymbol{\theta}_i, \boldsymbol{\theta}_j) &= \mathbb{E}_{\mathbf{x}}\big[\sigma(\boldsymbol{w}_i^\top\mathbf{x} + b_i)\sigma(\boldsymbol{w}_j^\top\mathbf{x} + b_j)\big] \\
&= \mathbb{E}_{\mathbf{z}}\big[\sigma(\boldsymbol{w}_i^\top\boldsymbol{A}\mathbf{z} + \boldsymbol{w}_i^\top\boldsymbol{m} + b_i)\sigma(\boldsymbol{w}_j^\top\boldsymbol{A}\mathbf{z} + \boldsymbol{w}_j^\top\boldsymbol{m} + b_j)\big] \\
&= k_{\sigma, \mathrm{Id}, \Phi}(\mathcal{T}\boldsymbol{\theta}_i, \mathcal{T}\boldsymbol{\theta}_j).
\end{aligned}$$

$\qquad\square$

**Kernel 2.** $k_{\cos,\mathrm{Id},\Phi}(\boldsymbol{\theta}_i,\boldsymbol{\theta}_j) = \frac{\cos|b_i-b_j|}{2}\exp\big(\frac{-\|\boldsymbol{w}_j-\boldsymbol{w}_j\|^2}{2}\big) + \frac{\cos|b_i+b_j|}{2}\exp\big(\frac{-\|\boldsymbol{w}_j+\boldsymbol{w}_j\|^2}{2}\big).$

*Proof.* First observe that the expected value of the cosine of a Gaussian random variable can be evaluated by equating the real and imaginary components of the characteristic function of a Gaussian random variable and the expected value of Euler's form. That is, if $z$ is Gaussian with mean $m$ and variance $v^2$,

$$\mathbb{E}e^{iz} = \mathbb{E}[\cos(z)] + i\mathbb{E}[\sin(z)] = e^{im-\frac{1}{2}v^2}$$
$$= (\cos\mu + i\sin m)e^{-\frac{1}{2}v^2}$$
$$\implies \mathbb{E}[\cos(z)] = \cos(m)e^{-\frac{1}{2}v^2}.$$

With this identity at hand, we proceed by direct evaluation of (4).

$$k_{\cos,\mathrm{Id},\Phi}(\boldsymbol{\theta}_i,\boldsymbol{\theta}_j) = \mathbb{E}_{\mathbf{x}}\big[\cos(\boldsymbol{w}_i^\top\mathbf{x}+b_i)\cos(\boldsymbol{w}_j^\top\mathbf{x}+b_j)\big], \quad \mathbf{x}\sim\mathcal{N}(\mathbf{0},\boldsymbol{I})$$
$$= \frac{1}{2}\mathbb{E}_{\mathbf{x}}\big[\cos\big((\boldsymbol{w}_i-\boldsymbol{w}_j)^\top\mathbf{x}+(b_i-b_j)\big) + \cos\big((\boldsymbol{w}_i+\boldsymbol{w}_j)^\top\mathbf{x}+(b_i+b_j)\big)\big]$$
$$= \frac{1}{2}\cos|b_i-b_j|\exp\big(-\frac{1}{2}\|\boldsymbol{w}_i-\boldsymbol{w}_j\|^2\big) + \cos|b_i+b_j|\exp\big(-\frac{1}{2}\|\boldsymbol{w}_i+\boldsymbol{w}_j\|^2\big).$$
$\square$

**Kernel 4.** $k_{\mathrm{Id},\mathrm{Id},\Phi}(\boldsymbol{\theta}_i,\boldsymbol{\theta}_j) = \boldsymbol{w}_i^\top\boldsymbol{w}_j + b_i b_j.$

*Proof.* This is immediate from the expected value of a product of two correlated Gaussians, $\boldsymbol{w}_i^\top\mathbf{x}+b_i$ and $\boldsymbol{w}_j^\top\mathbf{x}+b_j$. $\square$

**Kernel 5.** *The kernel* $k_{\mathrm{Snake}_a(\cdot)-\frac{1}{2a},\mathrm{Id},\Phi}(\boldsymbol{\theta}_i,\boldsymbol{\theta}_j)$ *is equal to*

$$\frac{1}{4a^2}k_{\cos,\mathrm{Id},\Phi}(2a\boldsymbol{\theta}_i,2a\boldsymbol{\theta}_j) + \boldsymbol{w}_j^\top\boldsymbol{w}_j\Big(\sin(2ab_j)e^{-2a^2\|\boldsymbol{w}_j\|^2} + \sin(2ab_i)e^{-2a^2\|\boldsymbol{w}_i\|^2}\Big)$$
$$-\frac{b_i}{2a}\cos(2ab_j)e^{-2a^2\|\boldsymbol{w}_j\|^2} - \frac{b_j}{2a}\cos(2ab_i)e^{-2a^2\|\boldsymbol{w}_i\|^2} + k_{\mathrm{Id},\mathrm{Id},\Phi}(\boldsymbol{\theta}_i^{(1)},\boldsymbol{\theta}_j^{(1)}).$$

*Proof.* Choosing $\sigma = \mathrm{Snake}_a(\cdot) - \frac{1}{2a}$ in (4), and expanding the resulting quadratic,

$$k_{\mathrm{Snake}_a(\cdot)-\frac{1}{2a},\mathrm{Id}}(\boldsymbol{\theta}_i,\boldsymbol{\theta}_j)$$
$$= \underbrace{\frac{1}{4a^2}\mathbb{E}_{\mathbf{x}}\big[\cos\big(2a(\boldsymbol{w}_i^\top\mathbf{x}+b_i)\big)\cos\big(2a(\boldsymbol{w}_j^\top\mathbf{x}+b_j)\big)\big]}_{\text{Kernel 2}} - \underbrace{\frac{1}{2a}\mathbb{E}_{\mathbf{x}}\big[(\boldsymbol{w}_i^\top\mathbf{x}+b_i)\cos\big(2a(\boldsymbol{w}_j^\top\mathbf{x}+b_j)\big)\big]}_{\triangleq A} -$$
$$\underbrace{\frac{1}{2a}\mathbb{E}_{\mathbf{x}}\big[\cos\big(2a(\boldsymbol{w}_i^\top\mathbf{x}+b_i)\big)(\boldsymbol{w}_j^\top\mathbf{x}+b_j)\big]}_{\triangleq B} + \underbrace{\mathbb{E}_{\mathbf{x}}\big[(\boldsymbol{w}_i^\top\mathbf{x}+b_i)(\boldsymbol{w}_j^\top\mathbf{x}+b_j)\big]}_{\text{Kernel 4}}. \tag{15}$$

We now evaluate $A$ and $B$. The terms $A$ and $B$ obey a symmetry, so it suffices to evaluate term $A$. Term $A$ can be evaluated using Stein's lemma,

$$A = \frac{1}{2a}\mathbb{E}\big[(z_1+b_i)\cos(z_2+2ab_j)\big], \quad (z_1,z_2)^\top \sim \mathcal{N}\left(\begin{pmatrix}0\\0\end{pmatrix}, \begin{pmatrix}\boldsymbol{w}_i^\top\boldsymbol{w}_i & 2a\boldsymbol{w}_i^\top\boldsymbol{w}_j\\ 2a\boldsymbol{w}_j^\top\boldsymbol{w}_i & 4a^2\boldsymbol{w}_j^\top\boldsymbol{w}_j\end{pmatrix}\right)$$
$$= \frac{1}{2a}\mathbb{E}\big[z_1\cos(z_2+2ab_j)\big] + \frac{b_i}{2a}\mathbb{E}\big[\cos(z_2+2ab_j)\big]$$
$$= -\boldsymbol{w}_i^\top\boldsymbol{w}_j\mathbb{E}\big[\sin(z_2+2ab_j)\big] + \frac{b_i}{2a}\mathbb{E}\big[\cos(z_2+2ab_j)\big]$$
$$= -\boldsymbol{w}_i^\top\boldsymbol{w}_j\sin(2ab_j)\exp(-2a^2\|\boldsymbol{w}_j\|^2) + \frac{b_i}{2a}\cos(2ab_j)\exp(-2a^2\|\boldsymbol{w}_j\|^2).$$

Assembling all the known individual terms in (15),

$$k_{\mathrm{Snake}_a(\cdot)-\frac{1}{2a},\mathrm{Id}}(\boldsymbol{\theta}_i,\boldsymbol{\theta}_j)$$

$$= \frac{1}{4a^2}k_{\cos,\mathrm{Id}}(2a\boldsymbol{\theta}_i,2a\boldsymbol{\theta}_j) + \boldsymbol{w}_i^\top \boldsymbol{w}_j\Big(\sin(2ab_j)\exp(-2a^2\|\boldsymbol{w}_j\|^2) + \sin(2ab_i)\exp(-2a^2\|\boldsymbol{w}_i\|^2)\Big)$$

$$- \frac{b_i}{2a}\cos(2ab_j)\exp(-2a^2\|\boldsymbol{w}_j\|^2) - \frac{b_j}{2a}\cos(2ab_i)\exp(-2a^2\|\boldsymbol{w}_i\|^2) + k_{\mathrm{Id},\mathrm{Id}}(\boldsymbol{\theta}_i^{(1)},\boldsymbol{\theta}_j^{(1)}).$$

$\square$

**Kernel 6.** *The kernel* $k_{\mathrm{Snake}_a,\mathrm{Id},\Phi}(\boldsymbol{\theta}_i,\boldsymbol{\theta}_j)$ *is equal to*

$$\frac{1}{2a}\Big(b_i - \frac{1}{2a}\cos(2ab_i)\exp(-2a^2\|\boldsymbol{w}_i\|^2) + b_j - \frac{1}{2a}\cos(2ab_j)\exp(-2a^2\|\boldsymbol{w}_j\|^2)\Big)$$

$$+ k_{\mathrm{Snake}_a(\cdot)-\frac{1}{2a},\mathrm{Id},\Phi}(\boldsymbol{\theta}_i,\boldsymbol{\theta}_j) + \frac{1}{4a^2}.$$

*Proof.* Choose $\sigma = \mathrm{Snake}_a$ and note that $\mathrm{Snake}_a(\cdot) = \Big(\mathrm{Snake}_a(\cdot) - \frac{1}{2a}\Big) + \frac{1}{2a}$. The Kernel 5 corresponds with the case $\Big(\mathrm{Snake}_a(\cdot) - \frac{1}{2a}\Big)$, so we are left with three additional terms. These terms may be evaluated directly,

$$k_{\mathrm{Snake}_a,\mathrm{Id}}(\boldsymbol{\theta}_i,\boldsymbol{\theta}_j)$$

$$= k_{\mathrm{Snake}_a(\cdot)-\frac{1}{2a},\mathrm{Id}}(\boldsymbol{\theta}_i,\boldsymbol{\theta}_j) + \frac{1}{4a^2}+$$

$$\frac{1}{2a}\Big(b_i - \frac{1}{2a}\cos(2ab_i)\exp(-2a^2\|\boldsymbol{w}_i\|^2) + b_j - \frac{1}{2a}\cos(2ab_j)\exp(-2a^2\|\boldsymbol{w}_j\|^2)\Big).$$

$\square$

## C Examples which generalise standard exponential family models

In this section, we will study examples of the SNEFY model which use activation function $\sigma(u) = \exp(u/2)$ (or equivalently, up to scaling, $\sigma(u) = \exp(u)$) and as such correspond to a notion of exponential family mixture models allowing negative weights, as discussed in Section 3.2. These examples have tractable kernels whenever the corresponding exponential family has a tractable normalising constant and we can write the kernels directly using Proposition 1.

SNEFY **Von Mises-Fisher mixtures.** The VMF distribution is a helpful way of defining the notion of a Gaussian distribution to the sphere. The following kernel may be used to define a VMF distribution. Alternatively, it may be viewed as a way of constructing a distribution supported on $\mathbb{R}^d$ with sufficient statistics which are projected onto the sphere.

**Kernel 3.** *Define* $proj_{\mathbb{S}^{d-1}}(\boldsymbol{x}) \triangleq \boldsymbol{x}/\|\boldsymbol{x}\|$ *to be the projection onto the unit sphere. Then*

$$k_{\exp,proj_{\mathbb{S}^{d-1}},\Phi}(\boldsymbol{\theta}_i,\boldsymbol{\theta}_j) = \exp(b_i+b_j)\frac{\Gamma(d/2)2^{d/2-1}I_{d/2-1}\big(\|\boldsymbol{w}_i+\boldsymbol{w}_j\|\big)}{\|\boldsymbol{w}_i+\boldsymbol{w}_j\|^{d/2-1}},$$

*where* $I_p$ *is the modified Bessel function of the first kind of order* $p$. *In the special case* $d = 3$, *we have the closed-form* $k_{\exp,proj_{\mathbb{S}^2},\Phi}(\boldsymbol{\theta}_i,\boldsymbol{\theta}_j) = \exp(b_i+b_j)\frac{\big(e^{\|\boldsymbol{w}_i+\boldsymbol{w}_j\|}-e^{-\|\boldsymbol{w}_i+\boldsymbol{w}_j\|}\big)}{2\|\boldsymbol{w}_i+\boldsymbol{w}_j\|}.$

*Proof.* If $\mathbf{x} \sim \mathcal{N}(0, \boldsymbol{I})$, then $\mathbf{x}/\|\mathbf{x}\|$ is uniformly distributed on the sphere. From the normalizing constant of the von Mises-Fisher distribution, from Proposition 1, it then follows that

$$
\begin{aligned}
k_{\exp,\mathrm{proj}_{\mathbb{S}^{d-1}},\Phi}(\boldsymbol{\theta}_i, \boldsymbol{\theta}_j) &= \mathbb{E}_{\mathbf{x}}\big[\exp\big(\boldsymbol{w}_i^\top \mathbf{x}/\|\mathbf{x}\| + b_i + \boldsymbol{w}_j^\top \mathbf{x}/\|\mathbf{x}\| + b_j\big)\big], \quad \mathbf{x} \sim \mathcal{N}(\mathbf{0}, \boldsymbol{I}) \\
&= \exp(b_i + b_j) \int_{\mathbb{S}^{d-1}} \exp\big((\boldsymbol{w}_i + \boldsymbol{w}_j)^\top \boldsymbol{x}\big)\, d\boldsymbol{x} \frac{\Gamma(d/2)}{2\pi^{d/2}} \\
&= \frac{\exp(b_i + b_j)\Gamma(d/2)}{2\pi^{d/2}} \int_{\mathbb{S}^{d-1}} \exp\big(\|\boldsymbol{w}_i + \boldsymbol{w}_j\|\boldsymbol{a}^\top \boldsymbol{x}\big)\, d\boldsymbol{x}, \quad \text{where } \boldsymbol{a} \text{ is a unit vector} \\
&= \frac{\exp(b_i + b_j)\Gamma(d/2)}{2\pi^{d/2}} \frac{(2\pi)^{d/2} I_{d/2-1}\big(\|\boldsymbol{w}_i + \boldsymbol{w}_j\|\big)}{\|\boldsymbol{w}_i + \boldsymbol{w}_j\|^{d/2-1}} \\
&= \exp(b_i + b_j) \frac{\Gamma(d/2)2^{d/2-1} I_{d/2-1}\big(\|\boldsymbol{w}_i + \boldsymbol{w}_j\|\big)}{\|\boldsymbol{w}_i + \boldsymbol{w}_j\|^{d/2-1}},
\end{aligned}
$$

where $I_p$ is the modified Bessel function of the first kind of order $p$. In the special case of $p = 1/2$, we have $I_{1/2}(z) = \sqrt{\frac{2}{\pi z}}\sinh(z) = \big(\exp(z) - \exp(-z)\big)\sqrt{\frac{1}{2\pi z}}$. This implies that when $d = 3$, since $\Gamma(3/2) = \frac{\sqrt{\pi}}{2}$,

$$
k_{\exp,\mathrm{proj}_{\mathbb{S}^2},\Phi}(\boldsymbol{\theta}_i, \boldsymbol{\theta}_j) = \exp(b_i + b_j)\frac{\big(e^{\|\boldsymbol{w}_i + \boldsymbol{w}_j\|} - e^{-\|\boldsymbol{w}_i + \boldsymbol{w}_j\|}\big)}{2\,\|\boldsymbol{w}_i + \boldsymbol{w}_j\|}.
$$

$\square$

Note that $k_{\exp,\mathrm{proj}_{\mathbb{S}^{d-1}},\Phi}(\boldsymbol{\theta}_i, \boldsymbol{\theta}_j) = k_{\exp,\mathrm{Id},\nu}(\boldsymbol{\theta}_i, \boldsymbol{\theta}_j)$, where $\nu$ is the uniform measure on the sphere $\mathbb{S}^{d-1}$, because if $\mathbf{x}$ is Gaussian then $\mathbf{x}/\|\mathbf{x}\|$ is uniform on the sphere. This allows one to construct $\mathrm{SNEFY}_{\exp,\mathrm{Id},\nu}$ distributions, which are certain "mixtures" of VMF distributions with weights $\boldsymbol{V}^\top \boldsymbol{V}$.

SNEFY **Gaussian mixtures, fixed variance.** We may similarly construct kernels corresponding to "mixtures" of Gaussian distributions. The case here corresponds to a case of known fixed variance parameter.

**Kernel 7.** $k_{\exp,\mathrm{Id},\Phi}(\boldsymbol{\theta}_i, \boldsymbol{\theta}_j) = \exp(b_i + b_j)\exp\big(\frac{1}{2}\|\boldsymbol{w}_i + \boldsymbol{w}_j\|^2\big)$.

*Proof.* This is a consequence of the moment generating function of the multivariate Gaussian distribution. More concretely, by Proposition 1,

$$
\begin{aligned}
k_{\exp,\mathrm{Id},\Phi}(\boldsymbol{\theta}_i, \boldsymbol{\theta}_j) &= \mathbb{E}_{\mathbf{x}}\big[\exp\big(\boldsymbol{w}_i^\top \mathbf{x} + b_i + \boldsymbol{w}_j^\top \mathbf{x} + b_j\big)\big], \quad \mathbf{x} \sim \mathcal{N}(\mathbf{0}, \boldsymbol{I}) \\
&= \exp(b_i + b_j)\mathbb{E}_{\mathbf{x}}\big[\exp\big((\boldsymbol{w}_i + \boldsymbol{w}_j)^\top \mathbf{x}\big)\big] \\
&= \exp(b_i + b_j)\exp\big(\frac{1}{2}\|\boldsymbol{w}_i + \boldsymbol{w}_j\|^2\big).
\end{aligned}
$$

$\square$

SNEFY **Poisson mixtures.** Most of our examples deal with continuous distributions but in fact SNEFY can readily be used for discrete distribution modelling. This is particularly helpful when the support is large or infinite, for which computing normalising constants can naively be challenging even in the discrete setting. Let $\mathbb{X} = \{0, 1, 2, \ldots\}$, $t(x) = x$, and the base measure $\mu(dx) = 1/x!\,\nu(dx)$, where $\nu$ is the counting measure. A SNEFY model for a probability mass function which is a mixture of Poisson distributions allowing negative weights is given by

$$
\begin{aligned}
p(x; \boldsymbol{V}, \boldsymbol{w}) &= \frac{1}{\mathrm{Tr}\,(\boldsymbol{V}^\top \boldsymbol{V}\boldsymbol{K}_\Theta)}\frac{1}{x!}\sum_{i=1}^n \sum_{j=1}^n \boldsymbol{v}_{.,i}^\top \boldsymbol{v}_{.,j} \exp\big((w_i + w_j)x\big) \\
&= \frac{1}{\mathrm{Tr}\,(\boldsymbol{V}^\top \boldsymbol{V}\boldsymbol{K}_\Theta)}\frac{1}{x!}\sum_{i=1}^n \sum_{j=1}^n \boldsymbol{v}_{.,i}^\top \boldsymbol{v}_{.,j}(\lambda_i\lambda_j)^x, \quad x = 0, 1, 2, \ldots
\end{aligned}
$$

following the usual mean parametrisation $\lambda_i = e^{w_i}$, so the individual mixture components have rates which are geometric means of $(\lambda_i, \lambda_j)$ pairs.

**Kernel 8.** *Choose the base measure $\mu(dx) = (x!)^{-1} \nu(dx)$, where $\nu$ is the counting measure. We have*

$$k_{\exp,\mathrm{Id},(x!)^{-1}\nu}(\boldsymbol{\theta}_i, \boldsymbol{\theta}_j) = \exp\left(b_i + b_j\right) \exp\left(\exp(w_i + w_j)\right).$$

*Proof.* This is again direct from Proposition 1. In detail,

$$k_{\exp,\mathrm{Id},(x!)^{-1}\nu}(\boldsymbol{\theta}_i, \boldsymbol{\theta}_j) = \sum_{x=0}^{\infty} \frac{1}{x!} \exp\left(w_i x + w_j x + b_i + b_j\right)$$

$$= \exp\left(b_i + b_j\right) \sum_{x=0}^{\infty} \frac{1}{x!} \exp\left(w_i x + w_j x\right)$$

The second factor involving the sum is the partition function of the Poisson distribution in canonical form, which is $\exp\left(\exp(w_i + w_j)\right)$. $\qquad\square$

The usual mean parameterisation of the Poisson distribution is through a rate parameter $\lambda_i = \exp(w_i)$, which would lead to the kernel representation

$$k_{\exp,\mathrm{Id},(x!)^{-1}\nu}(\boldsymbol{\theta}_i, \boldsymbol{\theta}_j) = \exp\left(b_i + b_j\right) \exp\left(\lambda_i \lambda_j\right).$$

SNEFY **Gaussian mixtures, unknown variance (Squared radial basis function network).**  We now discuss an intriguing connection between the Gaussian distribution and squared RBF networks. This connection is made possible through our machinery of SNEFY distributions. Let $\mathbb{X} = \mathbb{R}^d$, $\boldsymbol{t}(\boldsymbol{x}) = (x_1, \ldots, x_d, x_1^2, \ldots, x_d^2)$ (i.e. $D = 2d$) and suppose $\mu(dx) = d\boldsymbol{x}$ is Lebesgue measure. Choose $\sigma(\cdot) = \exp(\cdot/2)$ and consider the $r$-th output of our network $\boldsymbol{f} : \mathbb{R}^D \to \mathbb{R}^m$

$$f_r(\boldsymbol{t}(\boldsymbol{x}); \boldsymbol{V}, \boldsymbol{\Theta}) = \sum_{i=1}^{n} v_{ri} \exp\left(\frac{1}{2}\left(\sum_{\ell=1}^{d} w_{i\ell} x_\ell + \sum_{\ell=1}^{d} \tilde{w}_{i\ell} x_\ell^2\right)\right),$$

where we denoted $\tilde{w}_{i\ell} = w_{i,d+\ell}$. In this case, we require that $\tilde{w}_{e\ell} < 0$ for the model to be (square) integrable. Reparametrising $\sigma_{i\ell}^2 = -\frac{1}{2\tilde{w}_{i\ell}}$ and $\mu_{i\ell} = -\frac{w_{i\ell}}{2\tilde{w}_{i\ell}}$ and absorbing the factor $\exp\left(-\sum_{\ell=1}^{d} \frac{\mu_{i\ell}^2}{4\sigma_{i\ell}^2}\right)$ into readout parameters $\boldsymbol{V}$, gives

$$f_r(\boldsymbol{t}(\boldsymbol{x}); \boldsymbol{V}, \boldsymbol{\Theta}) = \sum_{i=1}^{n} v_{ri} \exp\left(-\sum_{\ell=1}^{d} \frac{(x_\ell - \mu_{i\ell})^2}{4\sigma_{i\ell}^2}\right).$$

Thus, we have recovered a classical radial basis function (RBF) network [52]. These models are well known to have universal approximation properties [36]. For the most commonly used form of the RBF network, we can restrict the parameters $\sigma_{i\ell}^2 = \sigma_i^2$ to be the same across the dimensions, giving

$$f_r(\boldsymbol{t}(\boldsymbol{x}); \boldsymbol{V}, \boldsymbol{\Theta}) = \sum_{i=1}^{n} v_{ri} \exp\left(-\frac{\|\boldsymbol{x} - \boldsymbol{\mu}_i\|^2}{4\sigma_i^2}\right), \quad \boldsymbol{x} \in \mathbb{R}^d,$$

with location parameters $\boldsymbol{\mu}_i \in \mathbb{R}^d$ and the scale parameters $\sigma_i^2 > 0$. Note the unusual factor of $4$ in front of $\sigma_i^2$ – this ensures that our model in fact reduces to the usual parametrisation of multivariate normal densities, since we will be modelling the density using the squared norm of $\boldsymbol{f}$.

Since $\mu$ is the Lebesgue measure, SNEFY gives us a density model with respect to the Lebesgue measure as

$$p(\boldsymbol{x}; \boldsymbol{V}, \boldsymbol{\Theta}) = \frac{1}{\mathrm{Tr}\left(\boldsymbol{V}^\top \boldsymbol{V} \boldsymbol{K}_{\boldsymbol{\Theta}}\right)} \sum_{i=1}^{n} \sum_{j=1}^{n} \boldsymbol{v}_{\cdot,i}^\top \boldsymbol{v}_{\cdot,j} \exp\left(-\frac{\|\boldsymbol{x} - \boldsymbol{\mu}_i\|^2}{4\sigma_i^2}\right) \exp\left(-\frac{\|\boldsymbol{x} - \boldsymbol{\mu}_j\|^2}{4\sigma_j^2}\right).$$

If $n = 1$, we recover simply a multivariate normal density $\mathcal{N}\left(\boldsymbol{\mu}_1, \sigma_1^2 I\right)$. The above model is essentially the same as the one in [42], despite being derived in a very different way.

**Kernel 9.** *Let $t^{(2)}(x) = (x_1, \ldots, x_d, x_1^2, \ldots, x_d^2)$ so that $D = 2d$ be the sufficient statistic. Partition $W = [W_{[:,1:d]}, \tilde{W}]$ and suppose $\tilde{W} < 0$ element-wise. Choose $\mu(dx) = dx$ to be the Lebesgue measure. Then*

$$k_{\exp,t^{(2)},dx}(\boldsymbol{\theta}_i, \boldsymbol{\theta}_j) = \pi^{d/2} \exp(b_i + b_j) \prod_{l=1}^{d} \exp\left(-\frac{(w_{il} + w_{jl})^2}{4(\tilde{w}_{il} + \tilde{w}_{jl})}\right) \frac{1}{\sqrt{-(\tilde{w}_{il} + \tilde{w}_{jl})}}$$

*Proof.* As with the kernels above, this follows from Proposition 1, since

$$k_{\exp,t^{(2)},dx}(\boldsymbol{\theta}_i, \boldsymbol{\theta}_j) = \int_{\mathbb{R}^d} \exp\left(\boldsymbol{w}_{i,1:d}^\top \boldsymbol{x} + \tilde{\boldsymbol{w}}_i^\top \boldsymbol{x}^2 + b_1 + \boldsymbol{w}_{j,1:d}^\top \boldsymbol{x} + \tilde{\boldsymbol{w}}_j^\top \boldsymbol{x}^2 + b_j\right) d\boldsymbol{x}$$

$$= (2\pi)^{d/2} \exp(b_i + b_j) \prod_{l=1}^{d} \exp\left(-\frac{(w_{il} + w_{jl})^2}{4(\tilde{w}_{il} + \tilde{w}_{jl})}\right) \frac{1}{\sqrt{-2(\tilde{w}_{il} + \tilde{w}_{jl})}}.$$

$\square$

While Proposition 1 gives us an expression for the kernel matrix $\boldsymbol{K}_\Theta$ in terms of natural parameters, we can also express it directly in terms of parameters $\boldsymbol{\mu}_i, \sigma_i^2$. In particular,

$$[\boldsymbol{K}_\Theta]_{ij} = \exp(b_i + b_j) \int_{\mathbb{R}^d} \exp\left(-\frac{\|\boldsymbol{x} - \boldsymbol{\mu}_i\|^2}{4\sigma_i^2}\right) \exp\left(-\frac{\|\boldsymbol{x} - \boldsymbol{\mu}_j\|^2}{4\sigma_j^2}\right) d\boldsymbol{x}$$

$$= \exp(b_i + b_j) \left(\frac{4\pi\sigma_i^2\sigma_j^2}{\sigma_i^2 + \sigma_j^2}\right)^{d/2} \exp\left(-\frac{\|\boldsymbol{\mu}_i - \boldsymbol{\mu}_j\|^2}{4\left(\sigma_i^2 + \sigma_j^2\right)}\right).$$

We briefly state two more cases without an extended discussion.

SNEFY **Gamma mixtures.**

**Kernel 10.** *Let $\mathbb{X} = (0, \infty)$, $t(x) = (\log x, -x)$ and $\sigma = \exp$. Partition $W = [W_{[:,1:d]}, \tilde{W}]$ and suppose $W_{[:,1:d]} > -1$ and $\tilde{W} > 0$ element-wise. Choose $\mu(dx) = dx$ to be the Lebesgue measure. Then*

$$k_{\exp,t,dx}(\boldsymbol{\theta}_i, \boldsymbol{\theta}_j) = \exp(b_i + b_j) \frac{\Gamma(w_{i1} + w_{j1} + 1)}{(w_{i2} + w_{j2})^{w_{i1} + w_{j1} + 1}}.$$

SNEFY **Dirichlet mixtures.**

**Kernel 11.** *Let $\mathbb{X} = \Delta^{D-1}$, a $(D-1)$-simplex of probability distributions, i.e. $\boldsymbol{x} \in [0, 1]^D$, $\sum_{i=1}^{D} x_i = 1$. Let $\sigma = \exp$. Let $t(\boldsymbol{x}) = (\log x_1, \ldots, \log x_D)$. Choose $\mu(d\boldsymbol{x}) = d\boldsymbol{x}$ to be the Lebesgue measure. Suppose $W > -1$ elementwise. Then*

$$k_{\exp,t,d\boldsymbol{x}}(\boldsymbol{\theta}_i, \boldsymbol{\theta}_j) = \exp(b_i + b_j) \frac{\prod_{d=1}^{D} \Gamma(w_{id} + w_{jd} + 1)}{\Gamma\left(D + \sum_{d=1}^{D} w_{id} + w_{jd}\right)}.$$

# D  Marginalisation in the case $\sigma = \exp(\cdot/2)$

Let $M$ be a positive semi-definite $n \times n$ matrix. We will make use of the following SNEFY parametrisation

$$P(d\boldsymbol{x}; \boldsymbol{M}, \Theta) = \frac{1}{z(\boldsymbol{M}, \Theta)} \sigma\left(\boldsymbol{W}t(\boldsymbol{x}) + \boldsymbol{b}\right)^\top \boldsymbol{M} \sigma\left(\boldsymbol{W}t(\boldsymbol{x}) + \boldsymbol{b}\right) \mu(d\boldsymbol{x}). \tag{16}$$

Since we can always write $\boldsymbol{M} = \boldsymbol{V}^\top \boldsymbol{V}$, for an $m \times n$ matrix $\boldsymbol{V}$, $m \leq n$, we have

$$\sigma\left(\boldsymbol{W}t(\boldsymbol{x}) + \boldsymbol{b}\right)^\top \boldsymbol{M} \sigma\left(\boldsymbol{W}t(\boldsymbol{x}) + \boldsymbol{b}\right) = \|\boldsymbol{V}\sigma\left(\boldsymbol{W}t(\boldsymbol{x}) + \boldsymbol{b}\right)\|^2 = \sum_{i=1}^{m}\left(\sum_{j=1}^{n} v_{ij}\sigma\left(\boldsymbol{w}_j^\top t(\boldsymbol{x}) + b_j\right)\right)^2,$$

which is, as in the parametrisation given in the main text, simply the squared Euclidean norm of a multi-output neural network, with $\boldsymbol{V}$ corresponding to the weights of the second layer. If we denote by $\boldsymbol{v}_{\cdot j}$ the $j$-th column of $\boldsymbol{V}$, the normalizing constant is given by

$$\sum_{j=1}^{n}\sum_{l=1}^{n}\boldsymbol{v}_{\cdot j}^{\top}\boldsymbol{v}_{\cdot l}k_{\sigma,\boldsymbol{t},\mu}(\boldsymbol{\theta}_j,\boldsymbol{\theta}_l) = \sum_{j=1}^{n}\sum_{l=1}^{n}m_{jl}k_{\sigma,\boldsymbol{t},\mu}(\boldsymbol{\theta}_j,\boldsymbol{\theta}_l) = \mathrm{Tr}(\boldsymbol{M}\boldsymbol{K}_{\boldsymbol{\Theta}})$$

where as before

$$k_{\sigma,\boldsymbol{t},\mu}(\boldsymbol{\theta}_i,\boldsymbol{\theta}_j) = \int \sigma\big(\boldsymbol{w}_i^{\top}\boldsymbol{t}(\boldsymbol{x}) + b_i\big)\sigma\big(\boldsymbol{w}_j^{\top}\boldsymbol{t}(\boldsymbol{x}) + b_j\big)\mu(d\boldsymbol{x}).$$

Now if we let $\sigma = \exp(\cdot/2)$ we obtain a family which is also closed under marginalisation (in addition to conditioning). The Proposition below generalises Proposition 1 of [42], which considers the special case of the Gaussian PSD mixtures.

**Proposition 3.** *Let* $\mathsf{x} = (\mathsf{x}_1, \mathsf{x}_2)$ *be jointly* $\mathtt{SNEFY}_{\mathbb{X}_1 \times \mathbb{X}_2, \boldsymbol{t}, \exp(\cdot/2), \mu}$ *with parameters* $\boldsymbol{V}$ *and* $\boldsymbol{\Theta} = ([\boldsymbol{W_1}, \boldsymbol{W_2}], \boldsymbol{b})$. *Assume that* $\mu(d\boldsymbol{x}) = \mu_1(d\boldsymbol{x}_1)\mu_2(d\boldsymbol{x}_2)$ *and* $\boldsymbol{t}(\boldsymbol{x}) = \big(\boldsymbol{t}_1(\boldsymbol{x}_1), \boldsymbol{t}_2(\boldsymbol{x}_2)\big)$. *Then the marginal distribution of* $\mathsf{x}_1$ *is* $\mathtt{SNEFY}_{\mathbb{X}_1, \boldsymbol{t}_1, \exp(\cdot/2), \mu_1}$ *with parameters* $\tilde{\boldsymbol{V}}$ *and* $\boldsymbol{\Theta} = (\boldsymbol{W_1}, \boldsymbol{b})$, *for some matrix* $\tilde{\boldsymbol{V}} \in \mathbb{R}^{m \times n}$.

*Proof.* Proposition 1 gives us the normalising constant for the parametrisation where biases are absorbed into $\boldsymbol{V}$. If we explicitly keep the biases in the parametrisation, we have

$$k_{\exp(\cdot/2),\boldsymbol{t},\mu}(\boldsymbol{\theta}_i,\boldsymbol{\theta}_j) = \exp\left(\frac{1}{2}(b_i + b_j)\right)\mathrm{z}_e\left(\frac{1}{2}(\boldsymbol{w}_i + \boldsymbol{w}_j)\right), \tag{17}$$

where $\mathrm{z}_e$ is the normalizing constant of the exponential family with the sufficient statistic $\boldsymbol{t}$ and base measure $\mu$. By Theorem 2, we have that

$$P_1(d\boldsymbol{x}_1; \boldsymbol{M}, \boldsymbol{\Theta}) = \frac{\mathrm{Tr}(\boldsymbol{M}\boldsymbol{C}_{\boldsymbol{\Theta}}(\boldsymbol{x}_1))}{\mathrm{Tr}(\boldsymbol{M}\boldsymbol{K}_{\boldsymbol{\Theta}})}\mu_1(d\boldsymbol{x}_1),$$

where $[\boldsymbol{C}_{\boldsymbol{\Theta}}(\boldsymbol{x}_1)]_{ij} = k_{\sigma,\boldsymbol{t}_2,\mu_2}\big((\boldsymbol{w}_{2i}, \boldsymbol{w}_{1i}^{\top}\boldsymbol{t}_1(\boldsymbol{x}_1) + b_i), (\boldsymbol{w}_{2j}, \boldsymbol{w}_{1j}^{\top}\boldsymbol{t}_1(\boldsymbol{x}_1) + b_j)\big)$. But now since $\sigma = \exp(\cdot/2)$, applying (17) gives

$$[\boldsymbol{C}_{\boldsymbol{\Theta}}(\boldsymbol{x}_1)]_{ij} = \mathrm{z}_{e,2}\left(\frac{1}{2}(\boldsymbol{w}_{2i} + \boldsymbol{w}_{2j})\right)\exp\left(\frac{1}{2}(\boldsymbol{w}_{1i}^{\top}\boldsymbol{t}_1(\boldsymbol{x}_1) + b_i)\right)\exp\left(\frac{1}{2}(\boldsymbol{w}_{1j}^{\top}\boldsymbol{t}_1(\boldsymbol{x}_1) + b_j)\right),$$

where $\mathrm{z}_{e,2}$ is the normalizing constant of the exponential family with the sufficient statistic $\boldsymbol{t}_2$ and base measure $\mu_2$. Thus, we can write

$$\mathrm{Tr}(\boldsymbol{M}\boldsymbol{C}_{\boldsymbol{\Theta}}(\boldsymbol{x}_1)) = \sum_{i=1}^{n}\sum_{j=1}^{n}\left\{m_{ij}\,\mathrm{z}_{e,2}\left(\frac{1}{2}(\boldsymbol{w}_{2i} + \boldsymbol{w}_{2j})\right)\right.$$

$$\left. \cdot \exp\left(\frac{1}{2}(\boldsymbol{w}_{1i}^{\top}\boldsymbol{t}_1(\boldsymbol{x}_1) + b_i)\right)\exp\left(\frac{1}{2}(\boldsymbol{w}_{1j}^{\top}\boldsymbol{t}_1(\boldsymbol{x}_1) + b_j)\right)\right\}$$

$$= \exp\left(\frac{1}{2}(\boldsymbol{W}_1^{\top}\boldsymbol{t}_1(\boldsymbol{x}_1) + \boldsymbol{b})\right)^{\top}\tilde{\boldsymbol{M}}\exp\left(\frac{1}{2}(\boldsymbol{W}_1^{\top}\boldsymbol{t}_1(\boldsymbol{x}_1) + \boldsymbol{b})\right)$$

and we conclude that the marginal is in the same family with $\tilde{\boldsymbol{M}} = \boldsymbol{M} \circ \boldsymbol{Z}_{e,2}$, where

$$[\boldsymbol{Z}_{e,2}]_{i,j} = \mathrm{z}_{e,2}\left(\frac{1}{2}(\boldsymbol{w}_{2i} + \boldsymbol{w}_{2j})\right).$$

Note that $\tilde{\boldsymbol{M}}$ is PSD as an Hadamard product of two PSD matrices. Thus, we can find $\tilde{\boldsymbol{V}}$ such that $\tilde{\boldsymbol{M}} = \tilde{\boldsymbol{V}}^{\top}\tilde{\boldsymbol{V}}$. $\qquad\square$

|        | SNEFY 1 | Resampled 1 | Gauss 1 | GMM 1 |
|--------|---------|-------------|---------|-------|
| Moons | $-1.59 \pm 0.03$ | $-1.76 \pm 0.02$ | $-3.29 \pm 0.01$ | $-1.57 \pm 0.03$ |
|       | 1431 | 68007 | 1194 | 1240 |
|       | $1352.19 \pm 134.30$ | $615.57 \pm 24.35$ | $730.21 \pm 23.60$ | $753.16 \pm 31.41$ |
| Circles | $-1.92 \pm 0.02$ | $-1.94 \pm 0.03$ | $-3.37 \pm 0.01$ | $-2.00 \pm 0.06$ |
|       | 1431 | 68007 | 1194 | 1240 |
|       | $798.13 \pm 196.89$ | $291.03 \pm 29.46$ | $162.64 \pm 17.19$ | $186.65 \pm 21.10$ |
| Rings | $-2.35 \pm 0.04$ | $-2.31 \pm 0.02$ | $-3.16 \pm 0.01$ | $-2.43 \pm 0.04$ |
|       | 1431 | 68007 | 1194 | 1240 |
|       | $1134.90 \pm 133.10$ | $502.32 \pm 24.51$ | $528.50 \pm 24.51$ | $559.12 \pm 30.90$ |

Table 4: As in Table 2, but with 1 NVP layer.

|        | SNEFY 2 | Resampled 2 | Gauss 2 | GMM 2 |
|--------|---------|-------------|---------|-------|
| Moons | $-1.58 \pm 0.02$ | $-1.59 \pm 0.03$ | $-1.62 \pm 0.03$ | $-1.58 \pm 0.02$ |
|       | 2621 | 69197 | 2384 | 2430 |
|       | $1453.05 \pm 147.67$ | $700.98 \pm 32.94$ | $831.40 \pm 49.24$ | $855.04 \pm 53.01$ |
| Circles | $-1.91 \pm 0.04$ | $-2.07 \pm 0.04$ | $-2.66 \pm 0.05$ | $-1.96 \pm 0.04$ |
|       | 2621 | 69197 | 2384 | 2430 |
|       | $899.77 \pm 196.54$ | $363.91 \pm 36.62$ | $267.23 \pm 32.05$ | $289.96 \pm 33.71$ |
| Rings | $-2.35 \pm 0.04$ | $-2.32 \pm 0.03$ | $-2.80 \pm 0.02$ | $-2.36 \pm 0.03$ |
|       | 2621 | 69197 | 2384 | 2430 |
|       | $1255.92 \pm 168.80$ | $579.02 \pm 23.70$ | $637.73 \pm 34.04$ | $658.98 \pm 37.69$ |

Table 5: As in Table 2, but with 2 NVP layers.

# E   Experiments

## E.1   2D Unconditional density estimation

Our benchmarking protocol is slightly altered compared with [46]. Firstly, we measure performance over 20 random seeds instead of 1 fixed seed. We find that sometimes the variance over random seeds can be large (e.g. Resampled 0 on Circles). Secondly, rather than computing test performance at the last epoch of training, we follow the more standard procedure of returning the test performance evaluated at the epoch corresponding with the smallest validation performance. This validation/test monitoring results in substantial performance gains in all of the models, with no extra computational cost for SNEFY, Gauss and GMM. For Resampled, monitoring the validation performance to a high precision requires estimating the normalising constant to a high precision, which is computationally challenging. We therefore only check the validation performance every 100 epochs, and compute a high precision normalising constant if the validation performance is the lowest encountered so far. We train each models for a maximum of 20000 iterations (while monitoring validation performance). We use Adam with default hyperparameters and weight decay $10^{-3}$. The batch size is $2^{10}$.

We use a SNEFY with Gaussian mixture model base measure supported on $\mathbb{X} = \mathbb{R}^2$, identity sufficient statistic $t$ and activation function cos. The base measure consists of 8 mixture components, each with a diagonal covariance matrix. We use the same Resampled architecture as in the original paper [46]. We use an MLP with layer widths $[2, 256, 256, 1]$ and sigmoid activations for the resampling distribution. We use a discount factor of $0.1$ for the exponential moving average partition function calculation. Note that for Resampled models we are only able to provide an estimate of the test log likelihood, not the exact log likelihood as in other methods. We choose $n = 50$ and $m = 1$. The Gaussian mixture model has 10 mixture components, each with a diagonal covariance matrix. Each normalising flow block consists of an affine coupling block, a permutation layer, and an actnorm layer.

## E.2   Modelling distributions on the sphere

We compare the performance of a VMF distribution, a "regular" VMF mixture distribution and our SNEFY construction of the VMF mixture, which allows for some negative weight coefficients as discussed in § 3.2. We use the dataset [44] also used by [19] in a different context for point

|        | SNEFY 4 | Resampled 4 | Gauss 4 | GMM 4 |
|--------|---------|-------------|---------|-------|
| Moons  | $-1.58 \pm 0.03$ | $-1.60 \pm 0.07$ | $-1.60 \pm 0.03$ | $-1.57 \pm 0.03$ |
|        | 5001 | 71577 | 4764 | 4810 |
|        | $1616.21 \pm 152.61$ | $851.29 \pm 33.63$ | $1005.16 \pm 30.59$ | $1026.76 \pm 28.21$ |
| Circles| $-1.92 \pm 0.04$ | $-1.99 \pm 0.03$ | $-2.14 \pm 0.16$ | $-1.94 \pm 0.04$ |
|        | 5001 | 71577 | 4764 | 4810 |
|        | $1035.64 \pm 142.39$ | $514.02 \pm 47.98$ | $455.58 \pm 41.19$ | $480.15 \pm 44.61$ |
| Rings  | $-2.33 \pm 0.02$ | $-2.31 \pm 0.03$ | $-2.59 \pm 0.12$ | $-2.32 \pm 0.04$ |
|        | 5001 | 71577 | 4764 | 4810 |
|        | $1410.36 \pm 141.04$ | $749.10 \pm 52.05$ | $813.92 \pm 27.20$ | $833.36 \pm 30.77$ |

Table 6: As in Table 2, but with 4 NVP layers.

|        | SNEFY 8 | Resampled 8 | Gauss 8 | GMM 8 |
|--------|---------|-------------|---------|-------|
| Moons  | $-1.60 \pm 0.02$ | $-1.58 \pm 0.02$ | $-1.61 \pm 0.08$ | $-1.58 \pm 0.03$ |
|        | 9761 | 76337 | 9524 | 9570 |
|        | $2036.15 \pm 205.52$ | $1198.04 \pm 102.71$ | $1399.74 \pm 65.78$ | $1423.17 \pm 82.84$ |
| Circles| $-1.91 \pm 0.03$ | $-1.97 \pm 0.05$ | $-2.16 \pm 0.29$ | $-1.93 \pm 0.03$ |
|        | 9761 | 76337 | 9524 | 9570 |
|        | $1405.27 \pm 144.72$ | $839.23 \pm 92.99$ | $838.62 \pm 82.98$ | $862.25 \pm 81.09$ |
| Rings  | $-2.34 \pm 0.06$ | $-2.37 \pm 0.17$ | $-2.49 \pm 0.16$ | $-2.33 \pm 0.07$ |
|        | 9761 | 76337 | 9524 | 9570 |
|        | $1795.38 \pm 186.62$ | $1071.50 \pm 109.12$ | $1201.29 \pm 71.76$ | $1218.31 \pm 70.36$ |

Table 7: As in Table 2, but with 8 NVP layers.

processes. This dataset, retrieved in 2015, is called the "Revised New General Catalogue and Index Catalogue" (RNGC/IC). The RNGC/IC consists of locations of some $10610$ galaxies. We use the spherical coordinates of these galaxies and map them to the surface of a sphere.

We compare two types of $\text{SNEFY}_{\mathbb{S}^2,\exp,\mathrm{Id},d\boldsymbol{x}}$ with $m = n = 30$: one is constrained so that $\boldsymbol{V}$ is diagonal (and therefore $\boldsymbol{V}^\top \boldsymbol{V}$ is diagonal with nonnegative entries), and the other uses a unconstrained general $\boldsymbol{V}$. We expect the unconstrained model to be more expressive, and therefore obtain a better test NLL.

We use Adam with default hyperparameters and a full batch size. We randomly shuffle the data and perform an 80/20 train/test split. Each of the 50 runs uses a randomly sampled initialisation and train/test split. We train for $20000$ epochs.

### E.3 Conditional density estimation of photometric redshift

Our deep conditional feature extractor is an MLP that uses ReLU activations in each layer and batch normalisation between each layer. Our SNEFY model uses $\mathrm{Snake}_a$ activations with $a = 10$. The CNF models utilise affine layers wtih $\tanh$ activations. For SNEFY and CNF, we preprocess the input features so that they have sample mean zero and sample variance one. We train both deep learning models for 100 epochs using Adam with default hyperparameters and a batch size of 256. The dataset consists of 74309 training and 74557 examples, which are fixed between each run. Each run uses a randomly sampled initialisation (except for CKDE, which is deterministic).

## F  Complex activation case

Here we consider an extension where we allow the neural network activation $\sigma$ to be complex-valued, and accordingly the readout parameters $\boldsymbol{V}$ to be complex, i.e. $\boldsymbol{V} \in \mathbb{C}^{m \times n}$. Note that in this case

$$\left\| \boldsymbol{f}\left(\boldsymbol{t}(\boldsymbol{x}); \boldsymbol{V}, \boldsymbol{\Theta}\right) \right\|_2^2 \;=\; \left\| \boldsymbol{V}\sigma\left(\boldsymbol{W}\boldsymbol{t}(\boldsymbol{x}) + \boldsymbol{b}\right) \right\|^2 = \sum_{r=1}^m \left| \sum_{j=1}^n v_{rj}\sigma\left(\boldsymbol{w}_j^\top \boldsymbol{t}(\boldsymbol{x}) + b_j\right) \right|^2 .$$

Take $\sigma(u) = \exp(iu)$ and $\boldsymbol{t}(\boldsymbol{x}) = \boldsymbol{x}$.

Then the above takes the form

$$\sum_{r=1}^{m} \left| \sum_{j=1}^{n} v_{rj} \exp\left(i\left(\boldsymbol{w}_j^\top \boldsymbol{x} + b_j\right)\right) \right|^2 = \sum_{r=1}^{m} \sum_{j=1}^{n} \sum_{l=1}^{n} v_{rj} \bar{v}_{rl} e^{i\left(\boldsymbol{w}_j^\top \boldsymbol{x} + b_j\right)} e^{-i\left(\boldsymbol{w}_l^\top \boldsymbol{x} + b_l\right)}$$

$$= \sum_{r=1}^{m} \sum_{j=1}^{n} \sum_{l=1}^{n} v_{rj} \bar{v}_{rl} e^{i(b_j - b_l)} e^{i\left((\boldsymbol{w}_j - \boldsymbol{w}_l)^\top \boldsymbol{x}\right)}.$$

As in the $\sigma = \exp$ case, bias terms can be folded into the readout parameters $\boldsymbol{V}$ (which is the reason why we also require readout parameters to be complex). We note that the case $n = 1$ is not interesting as it simply reduces to the (normalised) base measure $\mu$.

In order to obtain the normalizing constant $\mathrm{Tr}(\boldsymbol{V}^H \boldsymbol{V} \boldsymbol{K_\Theta})$, we need the integral of the form

$$[\boldsymbol{K_\Theta}]_{jl} = k_{\exp(i\cdot),\mathrm{Id},\mu}(\boldsymbol{w}_j, \boldsymbol{w}_l) = \int e^{i\left((\boldsymbol{w}_j - \boldsymbol{w}_l)^\top \boldsymbol{x}\right)} \mu(d\boldsymbol{x}) =: \kappa(\boldsymbol{w}_j - \boldsymbol{w}_l), \qquad (18)$$

where $\kappa$ is simply the Fourier transform of the base measure $\mu$, i.e. its characteristic function in the case where $\mu$ is a probability measure. Hence many standard choices of $\mu$ lead to tractable normalizing constants. Note that while $\boldsymbol{V}$ and $\kappa$ both may be complex-valued, the normalizing constant as well as the density itself are real-valued.

Various examples of probability measures $\mu$ and its Fourier transforms that give rise to shift-invariant positive definite kernels $\kappa$ have been studied in the literature on Random Fourier Features (RFF) [38]. Here too, while the functional form of the expressions is identical, the role between the data and the parameters is reversed: in RFF, one is interested in approximating a given kernel on the data instances, by considering a probability measure on the frequency space which is that kernel's inverse Fourier transform.

**Remark 3.** *If we restrict our attention to real-valued $\boldsymbol{V}$ and thus explicitly maintain biases inside the parameterisation, this model can also be realised with stacking* cos *and* sin *activation so that they share the readout parameters since*

$$\sum_{r=1}^{m} \left| \sum_{j=1}^{n} v_{rj} \exp\left(i\left(\boldsymbol{w}_j^\top \boldsymbol{x} + b_j\right)\right) \right|^2 = \|\boldsymbol{V} \cos\left(\boldsymbol{W}\boldsymbol{x} + \boldsymbol{b}\right)\|^2 + \|\boldsymbol{V} \sin\left(\boldsymbol{W}\boldsymbol{x} + \boldsymbol{b}\right)\|^2.$$

## G  Discrete and mixed continuous/discrete SNEFYs

### G.1  Discrete SNEFYs via series

Closed-form expressions for normalising constants of discrete distributions are only advantageous if their computation costs less than naively summing up unnormalised density over all possible states. This is the case if the support has very large or infinite cardinality. An example that extends the classical Poisson distribution in exponential family form is given in Table 1. Here we discuss some other settings.

**NNK as a convergent series**  Suppose the support $\mathbb{X}$ is discrete and let $h(x)$ be a nonnegative function corresponding with the discrete base measure $\mu$. The NNK is given by

$$k_{\sigma,\boldsymbol{t},\mu}(\boldsymbol{\theta}_i, \boldsymbol{\theta}_j) = \sum_{\boldsymbol{x} \in \mathbb{X}} \sigma\left(\boldsymbol{w}_1^\top \boldsymbol{t}(\boldsymbol{x}) + b_1\right) \sigma\left(\boldsymbol{w}_2^\top \boldsymbol{t}(\boldsymbol{x}) + b_2\right) h(\boldsymbol{x}).$$

Such NNKs often resemble known convergent series.

**Fourier series**  For example, if $\sigma = \cos$ and $C_x = h(x)$ are some coefficients of a convergent Fourier series, then

$$k_{\sigma,\boldsymbol{t},\mu}(\boldsymbol{\theta}_i, \boldsymbol{\theta}_j) = \frac{1}{2} \sum_{x=0}^{\infty} C_x \Big( \cos\big(t(x)(w_1 - w_2) + (b_1 - b_2)\big) + \cos\big(t(x)(w_1 + w_2) + (b_1 + b_2)\big) \Big)$$

is just a series representation of a sum of two periodic functions. For example, if $C_x = \frac{2(-1)^{x+1}}{\pi x}$ for $x \geq 1$ and $C_x = 0$ otherwise, and $t(x) = 2\pi x$, the NNK is a sum of two sawtooth waves with frequencies $w_1 - w_2$ and $w_1 + w_2$ and phase offsets $b_1 - b_2$ and $b_1 + b_2$. Other examples include rectified sine waves, square waves and triangular waves. This extends to periodic functions with convergent multivariate Fourier series.

## G.2  Mixed continuous/discrete SNEFYs

Mixed distributions can be obtained by choosing the base measure to be a mixed continuous distribution. For example, choose $\mathbb{X} = \mathbb{R}^d$, $\mu(d\mathbf{x}) = \frac{1}{2}\big(\Phi(\mathbf{x}) + \delta(\mathbf{x})\big)\,d\mathbf{x}$, and take $\sigma$ and $\mathbf{t}$ to be any of the combinations leading to a closed-form NNGPK (for example, $\mathbf{t}$ as the identity and $\sigma$ as the error function, Leaky ReLU, GELU, cosine, or snake). Then

$$\begin{aligned}
k_{\sigma,\mathbf{t},\mu}(\theta_i, \theta_j) &= \frac{1}{2}\bigg( \int_{\mathbb{X}} \sigma\big(\mathbf{W}_i^\top \mathbf{t}(\mathbf{x}) + b_i\big)\sigma\big(\mathbf{W}_j^\top \mathbf{t}(\mathbf{x}) + b_j\big)\delta(\mathbf{x})d\mathbf{x} + \\
&\qquad \int_{\mathbb{X}} \sigma\big(\mathbf{W}_i^\top \mathbf{t}(\mathbf{x}) + b_i\big)\sigma\big(\mathbf{W}_j^\top \mathbf{t}(\mathbf{x}) + b_j\big)\Phi(\mathbf{x})d\mathbf{x}\bigg) \\
&= \frac{1}{2}\Big(\sigma\big(\mathbf{W}_i^\top \mathbf{t}(\mathbf{0}) + b_i\big)\sigma\big(\mathbf{W}_j^\top \mathbf{t}(\mathbf{0}) + b_j\big) + k_{\sigma,\mathbf{t},\Phi}(\theta_i, \theta_j)\Big),
\end{aligned}$$

which is a closed form.

