# OpenReview forum: "Squared Neural Families: A New Class of Tractable Density Models"
_NeurIPS.cc/2023/Conference — NeurIPS 2023 spotlight_

### Official Review · Reviewer_k4Y8 · 2023-06-13

**Soundness:** 3 good
**Presentation:** 2 fair
**Contribution:** 3 good
**Rating:** 5
**Confidence:** 3

**Summary:**

The paper introduces SNEFY, a tractable probabilistic model obtained by squaring the Euclidean norm of a neural net and normalizing it w.r.t. a base measure $\mu$ (cf. Equation 1).
The paper is mainly focused on studying settings in which one can obtain a closed form tractable solution for the normalizing constant of a given neural net.
These settings, which are listed in Table 1, are characterized by: a specific support, a base measure, a sufficient statistic, an activation function, bias/no bias and a kernel function (KNN).
For these settings, the normalization constant can be computed as a trace of a matrix which is quadratic in the number of neurons of the neural net (cf. Equation 6).
This matrix is a PSD matrix whose elements result from the application of a kernel function (NNK) over the hidden weights and biases of the neural net.
The model is trained by MLE and aveluated on: three 2D-dimensional datasets, a spheirical manifold and on a low-dimensional astronomy dataset.


**Strengths:**

- new NNK are proposed
- tractable marginalisation and conditioning
- connection with exponential families (every exponential famility is a SNEFY) and kernel-based methods for nonnegative functions


**Weaknesses:**

- SNEFY seems to be very slow to train and, moreover, performs very similar to GMMs when used as prior of a Normalizing Flow (see Tables 2-3, Appendix F2).
- Experiments are not really convincing and rather small scale. As a consequence, it is pretty unclear how expressive the model can actually be. Specifically, the 2D densities considered (see experiment section) are rather easy to fit. I'd suggest authors to investigate more complex 2D densities as the ones in [1].
- Sampling seems to be possible only by rejection sampling, and is therefore very computationally expensive. Could you elaborate more about sampling and its challenges?
- Minor: In general, I find that the exposition of the manuscript can be considerably improved. For instance, many equations/concepts are introduced more than once (compare contribution-background sections, Equation 1-2-3). Best results are not reported in bold, thus making Tables 2-3 not easy to read. There's no vspace between caption and Table 1. Table 3 is not referenced in the main text

[1] Nash, Charlie, and Conor Durkan. "Autoregressive energy machines." International Conference on Machine Learning. PMLR, 2019.

**Questions:**

- Why is "conditional SNEFY" (section 3.3) introduced before "marginal SNEFY" (section 3.4)? Doesn't tractable conditioning follow from tractable marginalisation ($P(A|B)=P(A \cap B) / P(B)$)?
- It is not clear to me whethere the method can be applied on discrete data. If so: (1) why didn't you include a  (simple) case study in your experiments? (2) Is it possible to handle both discrete and continuous features simultaneously?


**Limitations:**

I find authors to be open about the limitations of their model: They show that their model is slow to train (Tables 2-3) and that sampling is computationally expensive.
But probably, more emphasis and discussion about the long training time required would be appreciated.

---

> ### Author Rebuttal · Authors · 2023-08-04
>
> Our thanks for your thoughtful review and suggestions.
>
> **Weaknesses:**
> - " *Sampling seems to be possible only by rejection sampling, and is therefore very computationally expensive. Could you elaborate more about sampling?* "\
> There are some interesting approaches (beyond rejection sampling) and challenges associated with sampling. Please see our general response to reviewers for our comments on sampling.
> - " *SNEFY seems to be very slow to train and, moreover, performs very similar to GMMs when used as prior of a Normalizing Flow (see Tables 2-3, Appendix F2).* " \
> SNEFY (by itself, even without a normalising flow) outperforms the best GMM (which is allowed to have between 0 and 16 normalising flow layers) on Circles. It only takes on average 644 seconds, which is faster than the best performing GMM at at average of 862.25 seconds.
> - " *the 2D densities considered (see experiment section) are rather easy to fit. I'd suggest authors to investigate more complex 2D densities as the ones in [1].* "\
> We took a look at Einstein in [1]. In order to reproduce [1], 3 million training epochs will take roughly 20 of our GPU hours. Detailed experimentation on this data is unfortunately out of our budget at this stage. We will cite [1] as an EBM in the intro, where normalising constants are approximated rather than computed exactly.
>
> **Minor**
> - " *Best results are not reported in bold, thus making Tables 2-3 not easy to read. There's no vspace between caption and Table 1.* " \
> Thanks for your suggestions to improve the clarity of the tables. We will update these.
>
> - " *Table 3 is not referenced in the main text* " \
> Thanks for spotting this. We have updated the end of line 295 to refer to Table 3.
>
> **Questions:**
> - " *Why is "conditional SNEFY" (section 3.3) introduced before "marginal SNEFY" (section 3.4)? Doesn't tractable conditioning follow from tractable marginalisation?* "\
> We agree that tractable conditioning follows from tractable marginalisation, but the reverse also holds (by just rearranging Bayes' rule), which is the order of our proof. We don't see a strong reason to prefer one order over the other, however if you have reasons to preference one over the other, we are happy to swap the order according to your suggestions. Please let us know.
>
> - " *It is not clear to me whether the method can be applied on discrete data. If so: (1) why didn't you include a (simple) case study in your experiments? (2) Is it possible to handle both discrete and continuous features simultaneously?* " \
> Thanks for the great question. This made us think of some new special cases, which we will add to appendix. In detail:
>
> **Discrete and mixed continuous/discrete SNEFY:** \
> SNEFYs can indeed be applied to discrete data.
> Closed-form expressions for normalising constants of discrete distributions are only advantageous if their computation costs less than naively summing up unnormalised density over all possible states.
> This is the case if the support has very large or infinite cardinality.
> An example that extends the classical Poisson distribution in exponential family form is given in Table 1.
> Here we discuss some other settings.
>
> **NNK as a convergent series**
> Suppose the support $\mathbb{X}$ is discrete and let $h(\mathbf{x})$ be a nonnegative function corresponding with the discrete base measure $\mu$.
> The NNK is given by
> $$ k_{\sigma,\mathbf{t},\mu}(\mathbf{\theta}_i, \mathbf{\theta}_j) =  \sum   \sigma\big(\mathbf{w}_1^\top \mathbf{t}(\mathbf{x}) + b_1\big) \sigma\big(\mathbf{w}_2^\top \mathbf{t}(\mathbf{x}) + b_2\big) h(\mathbf{x}),$$
> where the sum runs over $\mathbf{x} \in \mathbb{X}$.
>
> Such NNKs often resemble known convergent series.
> For example, if $d=1$, $\sigma = \cos$ and $C_x = h(x)$ are some coefficients of a convergent Fourier series, then
> $$ k_{\sigma,\mathbf{t},\mu}(\mathbf{\theta}_i, \mathbf{\theta}_j) = \frac{1}{2} \sum  C_x\Big(  \cos \big(t(x) (w_1 - w_2) + (b_1-b_2) \big) + \cos \big(t(x) (w_1 + w_2) + (b_1+b_2)  \big) \Big) $$
> is just a series representation of a sum of two periodic functions.
> For example, if $C_x = \frac{2 (-1)^{x+1}}{\pi x}$ for $x \geq 1$ and $C_x = 0$ otherwise, and $t(x) = 2\pi x$, the NNK is a sum of two sawtooth waves with frequencies $w_1 - w_2$ and $w_1 + w_2$ and phase offsets $b_1 - b_2$ and $b_1 + b_2$.
> Other examples include rectified sine, square and triangular waves.
> This extends to functions with convergent multivariate Fourier series.
> We expect that the utility of such models is limited or difficult to uncover with benchmark datasets that we are aware of.
> If the cardinality of the discrete domain is small, discrete distributions can be flexibly represented by standard histograms, so utility of SNEFY parametrisation is unclear.
> In the SNEFY parameterisation, one biases the model away from general histograms, which may be favourable in terms of sample complexity if the data matches the bias or the discrete domain is very large, but otherwise not.
>
> **Mixed discrete/continuous:** \
> We did not originally think about mixed discrete/continuous distributions, but it is possible.
> We imagine such models may find more utility than the purely discrete case, although we have not looked into this further.
> Choose a mixed discrete/continuous base measure (e.g. a spike and slab Gaussian). Such induced NNKs will then be tractable in the same setting as the continuous component of the base measure is tractable, under extremely mild conditions on the activation function. E.g. if the base measure is a sum of a Gaussian measure and a Dirac delta, the resulting kernel is just a sum of the kernel with the Gaussian base measure and the product of the activations evaluated at $\mathbf{x} = \mathbf{0}.$ We will add this example to the appendix.

---

> > ### Comment · Reviewer_k4Y8 · 2023-08-10
> >
> > Thanks for your clarifications, and specifically for elaborating on discrete/mixed input variables. I did appreciate it.
> >
> > I read all reviews, and specifically the review from Reviewer zjNK, and I should say that I agree with basically everything they say: Sampling and limited evaluation (also due to long training times) are relevant weaknesses to me as well (as I had already pointed out).
> >
> > I like the work and will keep an eye on it but I'm still comfortable with my score.

---

> > > ### Author Response · Authors · 2023-08-13
> > >
> > > Thanks again for your feedback, especially regarding mixed and discrete settings, and confirming you have read the reviews and discussions.

---

### Official Review · Reviewer_zjNK · 2023-06-29

**Soundness:** 3 good
**Presentation:** 4 excellent
**Contribution:** 3 good
**Rating:** 6
**Confidence:** 4

**Summary:**

The authors introduce a class of normalized probability densities that are obtained by squaring a single-layer neural network. The class of densities, dubbed Squared Neural Family (SNEFY), offers a number of attractive properties including: i) they can be understood as a generalization of exponential family distributions: ii) normalizing constants can be computed in closed form; iii) they are closed under conditioning under certain factorization conditions; and iv) they are closed under marginalization in some special cases. The authors demonstrate that SNEFY can yield competitive log likelihoods on a few relatively simple density estimation tasks.

**Strengths:**

As I see it the main strengths of this submission include:
- It is very clearly written
- It addresses a problem (namely defining useful classes of density estimators) that is of wide interest in the NeurIPS community
- The proposed class of models is, to the best of my knowledge, novel
- The proposed class of models has some attractive properties, especially w.r.t. conditioning and marginalization
- The authors do a good job of placing their work in the context of related literature and methods



**Weaknesses:**

As I see it the main limitations and weaknesses of this submission are:
- The experimental evaluation is quite limited, making it difficult to judge the likely usefulness or impact of SNEFY in practice
- The intractability of obtaining samples from SNEFY distributions is a serious limitation (as the authors acknowledge)

To elaborate on the first point, the authors only consider three experiments, the first of which is in 2D. Indeed the only neural baseline appears to be a CNF described as follows in the appendix:
> The CNF models utilise affine layers wtih tanh activations.

Not only is this an inadequate description of this particular baseline, but the reader is left to wonder whether SNEFY models are actually competitive with more traditional normalizing flows in most density estimation tasks. Perhaps the joint learning of W and V in SNEFY doesn't work very well in practice because the parameterization is difficult to optimize and there's a tendency to get stuck in bad local optima? Even if SNEFY is not competitive with more traditional normalizing flow methods on common density estimation tasks, one might hope that its real value is to be found from leveraging some of its special properties, especially w.r.t. marginalization and conditioning. Unfortunately the experiments are not setup to demonstrate anything like this to the reader. The closest we have is the conditioning experiment involving astronomy data. While this is a good start, it is hardly convincing on its own. The authors note that "due to this tractability of the marginal distributions, it is straightforward to include the likelihood corresponding to incomplete observations"---unfortunately this experiment, which the reader anticipates with bated breath, is not forthcoming (similarly with the point process suggestion in the discussion section).

Granted, in its current form this paper makes some interesting mathematical and methodological contributions, but I believe a much stronger version of this paper would work harder to convince the reader that SNEFY might be more broadly useful. Otherwise one is left wondering whether SNEFY is just a mathematical curiosity, elegant in theory but not very practical or useful in real applications. It is of course not necessary to "beat" 100 baselines on 100 benchmarks, but I would love to see more evidence for the practical utility of SNEFY.

**Questions:**

- What are some of the practical trade-offs that you find empirically when using different values of m and n? E.g. do you see numerical instabilities or difficulties in optimization in certain regimes?
- What generic strategies might you recommend for choosing sufficient statistics `t(x)`? Is it generally necessary to learn these if you want a flexible model? If so does such joint learning tend to lead to difficulties in optimization? Or might it be sufficient to use random feature expansions or the like? One might suppose this might mitigate some optimization problems, should such problems occur.
- "For example, exploited..." [Line 29] This is a run-on (i.e. incomplete) sentence.
- Final line "... offering a model that scales like O(n^2) instead...". Do you mean O(n)?
- Dense tables like Table 2 are difficult to parse; consider making a plot

**Limitations:**

The main methodological limitation is the intractability of sampling, which the authors emphasize.

---

> ### Author Rebuttal · Authors · 2023-08-04
>
> Thank you for your in-depth review. We were very happy to read that you found that our model is novel, addresses a problem that is of wide interest to the community, admits attractive properties, and is well-placed in the context of related methods. We agree entirely with the limitation of sampling. Please see our "general response to reviewers" for more on this. Please see our comments below for responses to individual queries.
>
> **Weaknesses:**
> - *Further expeiments on partial observations* \
> Thanks for the excellent suggestion about experimenting with partial observations. We will add the following experiment. We consider the joint distribution of a random vector $\mathsf{x} \in \mathbb{R}^6$, with samples from the joint inputs and outputs of the astronomy dataset. The original training dataset is a matrix in $\mathbb{R}^{74309 \times 6}$. We partition this matrix into batches of size $256$ and randomly set each column of each batch to NaN with probability $p$ (i.e. the corresponding dimension is missing from observations). We then train a SNEFY model that utilises partial observations. We leave the original testing dataset, a matrix in $\mathbb{R}^{74557 \times 6}$, untouched, and measure the test NLL after training. We vary $p$ between $0.01$ and $1.0$ in steps of $0.01$, plotting the NLL as a function of $p$.  We also compare the performance of the SNEFY that uses partial observations with the performance of SNEFY that simply throws away incomplete observations, as well as the normalising flow baselines. Our observations are twofold: (1) Including partial observations increases performance and (2) Adding more complete observations increases performance. See attached figure in the general global rebuttal. We hope that the inclusion of such a demonstration does highlight some of the special properties of SNEFY, and satisfies your suggestion. Please let us know what you think.
>
> - *Poisson point processes, optimisation properties ...* \
> We agree that a detailed experimental evaluation of SNEFY would be interesting, especially teasing apart the novel aspects of the optimisation problem. We do hope to evaluate SNEFYs with respect to these new features in follow-up works, where we can properly dedicate enough room to subtle concepts. Poisson point processes are a very interesting avenue. In particular we have found that the form of the likelihood under the Poisson point process model leads to some optimisation problems with properties different from those of the likelihood maximization under an i.i.d. sampling from a density model. The synthetic experiments and real experiment on astronomy data were intended to show that in some special cases, SNEFYs are a competitive alternative to kernel density estimators, naive mixture models, and normalising flows.
>
>
> **Questions:**
> - "*What generic strategies might you recommend for choosing sufficient statistics $t(x)$? Is it generally necessary to learn these if you want a flexible model? If so does such joint learning tend to lead to difficulties in optimization? Or might it be sufficient to use random feature expansions or the like? One might suppose this might mitigate some optimization problems, should such problems occur.* " \
> As in the case for classical exponential families with closed form normalising constants, in this paper we take sufficient statistics to be known functions which allow for easy integration.
> So in some sense, the strategy is to use a table of known closed-forms (like table 1), and work with what you can use.
> Investigating whether there are interesting cases where $t$ itself is a neural network or a random feature expansion and the normalising constant is still tractable is another important avenue of further research. This is a technical challenge because for arbitrary $t$, it is not easy to compute the closed form normalising constants via the kernel as in table 1.
>
> - " *"For example, exploited..." [Line 29] This is a run-on (i.e. incomplete) sentence.* "  \
> Thanks for catching this typo. We will update this in the manuscript.
> - " *"Final line "... offering a model that scales like $\mathcal{O}(n^2)$ instead...".* "  \
> Thanks for catching this error. We meant to say, and will update the manuscript to say, "For example, we may build a Poisson point process intensity function using a squared neural network and compute the intensity function in closed-form, offering a model that scales quadratically in the number of neurons $\mathcal{O}(n^2)$ instead of comparable models which scale cubically in the number of datapoints $\mathcal{O}(N^3)$" [8, 45]."
>
> - " *Dense tables like Table 2 are difficult to parse; consider making a plot* " \
> Thanks for this suggestion. We will do this in the final version.

---

> > ### Comment · Reviewer_zjNK · 2023-08-10
> >
> > I thank the authors for their response, in particular the new experiment with partially observed data. While the empirical story is still a bit weak here, this goes some way in making the practical advantages of SNEFY more concrete. As such I will raise my score to a 6.
> >
> > I urge the authors to revise the way they present some of their empirical results: in particular the already-mentioned difficult-to-parse table but also the figure for the new experiment which is also difficult to parse.

---

> > > ### Author Response · Authors · 2023-08-13
> > >
> > > Thanks for your constructive feedback on our manuscript. We are glad that you urged us to consider the partial observation setting as an experiment. Regarding presentation of empirical results, we will add a figure for the tables and put in bold numbers for the final revision.

---

### Official Review · Reviewer_Y6Kj · 2023-07-06

**Soundness:** 3 good
**Presentation:** 4 excellent
**Contribution:** 2 fair
**Rating:** 7
**Confidence:** 3

**Summary:**

This paper introduces a parametric family of densities which admit tractable normalizing constants when a related neural-network kernel can be efficiently computed (e.g., in closed form).
The model is parameterized by the weights of a one-hidden-layer network, is closed under conditioning and admits tractable marginals that have a similar form as the global model.
The model is evaluated in unconditonal density modeling on synthetic data, and conditional density modeling on a redshift estimation task.

EDIT: I have read the author's rebuttal, which addressed my concerns.

**Strengths:**

The paper is very well-written and clear.
The proposed density model bears resemblance with the one considered in prior work [23, 38], but SNEFY is linear in the kernel as opposed to quadratic because it does not rely on a representer theorem.
It thus enjoys the tractability properties mentioned above.
I am not expert enough in this area to judge of the significance of the approach, but it seems interesting enough to me to be worth publishing.

**Weaknesses:**

The authors mention that the sufficient statistics $t_2$ are unrestricted in the unconditional case. However, it seems to me that this not a property of SNEFY per se but common to any conditional density model. For instance, one could consider a conditional GMM model where the means and variances of $x_1$ are general functions of $x_2$, which would then admit closed-form normalizing constants and marginalization.

**Questions:**

I did not understand the experiment setup in the unconditional density modeling experiments: how is the normalizing flow combined with the SNEFY model? It thought that the normalizing flow would play the role of the sufficient statistic $t$, which would prevent a closed-form computation of the kernel, but it seems not to be the case.

What is the relationship between considering a more advanced base density model (such as SNEFY rather than GMM) and more expressive normalizing flow? As a crude estimation, it might be fairer to compare a SNEFY model with a network of depth $L$ to a GMM model with a network of depth $L + 1$. In that case, the improvements of the SNEFY model are reduced. What does one gain by using a more expressive density model as opposed to a more expressive network?

**Limitations:**

The authors acknowledged the limitations of their approach: limited expressivity, and the general limitations of kernel-related methods. In particular, its complexity is cubic in the dimensionality (assuming $n$, $m$ and $d$ having similar orders of magnitude).

---

> ### Author Rebuttal · Authors · 2023-08-04
>
> Thank you for your kind review and useful feedback. In addition to the general rebuttal, we address your individual concerns below. Please let us know if you would like to discuss further.
>
> **Strengths:**
> - You are right to highlight a strength of our model is that it scales linearly in the number of datapoints, rather than other kernel based approaches which scale quadratically in the number of datapoints. We did not explicitly mention this in the paper (excluding the mention of this in the context of Poisson point processes at the end of the conclusion), so we will update the manuscript before publication.
>
> **Weaknesses:**
> - "*The authors mention that the sufficient statistics are unrestricted in the unconditional case. However, it seems to me that this not a property of SNEFY per se but common to any conditional density model. For instance, one could consider a conditional GMM model where the means and variances of $x_1$ are general functions of $x_2$, which would then admit closed-form normalizing constants and marginalization.*" \
> Thanks for bringing this up. You are right to say that a conditional GMM can have parameters that depend on the conditioning variable $x_2$ while still admitting a tractable normalising constant.
> Our claim was intended to be related to normalising flow models, not mixture models. Normalising flows in fact restrict dependence on the conditioning variable. In particular, in a conditional normalising flow (CNF), the mapping of each layer would have to be bijective in both $x_1$ and $x_2$ (see text above equation (7) in [48]). The point we were trying to make here is that CNFs (but not GMMs or other more flexible SNEFYs), the mapping is restricted to a bijection in both the variable of interest $x_1$ and the conditioning variable $x_2$, whereas we do not have this bijection requirement. We will update the text above section 5 to further clarify.
>
> **Questions:**
> - "*how is the normalizing flow combined with the SNEFY model? I thought that the normalizing flow would play the role of the sufficient statistic, which would prevent a closed-form computation of the kernel, but it seems not to be the case.*" \
> The SNEFY model is used as what is called the "base distribution" of the normalising flow (which is different to the base measure of SNEFY). For example, the most common base distribution in normalising flows is a Gaussian, so that a normalising flow likelihood represents the likelihood of a Gaussian random variable passed through a sequence of normalising flow layers. Instead of passing a Gaussian random variable through the normalising flow layers, we compute the likelihood of passing a SNEFY random variable through normalising flow layers. This allows combining the closed-form machinery of SNEFY with the closed-form machinery of normalising flows. We will update the experimental details appendix of the manuscript to clarify this point --- thanks for raising it.
> - " *What is the relationship between considering a more advanced base density model (such as SNEFY rather than GMM) and more expressive normalizing flow? As a crude estimation, it might be fairer to compare a SNEFY model with a network of depth to a GMM model with a network of depth . In that case, the improvements of the SNEFY model are reduced. What does one gain by using a more expressive density model as opposed to a more expressive network?* " \
> It is difficult to answer your first question mathematically, however we did try to investigate this empirically. For example, in table 3, we compare a zero layer normalising flow with SNEFY base (i.e. just a SNEFY) against normalising flows of various depths and Gaussian base distributions. We find that in this case, SNEFY outperforms normalising flows. As another 3 examples, compare table 2 and table 4 to tables 5 to 7. We see that a SNEFY with zero or 1 normalising flow layers appended to it is competitive with normalising flows with more layers, although a resampled base layer does beat SNEFY on Rings.

---

> > ### Comment · Reviewer_Y6Kj · 2023-08-11
> >
> > Thank you for their detailed answer.
> >
> > - I still disagree with the claim that "in a conditional normalising flow (CNF), the mapping of each layer would have to be bijective in both $x_1$ and $x_2$". Indeed, for maximum-likelihood training, one needs to compute the determinant of $\partial_{x_1} f(x_1 | x_2)$ only, so that the network should be invertible in $x_1$ for every $x_2$. In fact, in [48], the authors "the conditioning term
> > $x$ [(standing for $x_2$ in this paper's notations)] is transformed into a rich representation $h = g(x)$ using a large network $g$ [which does not need to be invertible]" (last paragraph of section 3.1). Am I missing something here?
> >
> > - I like the new missing data experiment, which showcases the usefulness of tractable marginalization in practical applications. To me, this is the strongest point of approach. I thus recommend acceptance.

---

> > > ### Author Response · Authors · 2023-08-13
> > > **Thanks for following up**
> > >
> > > You're right about point 1. The paragraph on line 295 is about bijectivity in the variable we are modelling, not the variable we are conditioning on. The last sentence of this paragraph is confusing because, as you point out, conditional normalising flows are not required to be bijective in the variable they are conditioning on. We should have been talking about the variable we are modelling, in which case normalising flows are restricted. We will update this paragraph in the final manuscript.
> > >
> > > We are glad that you like the new experiment. Thanks again for your feedback on the paper!

---

### Official Review · Reviewer_tpyH · 2023-07-10

**Soundness:** 4 excellent
**Presentation:** 3 good
**Contribution:** 3 good
**Rating:** 7
**Confidence:** 2

**Summary:**

The authors present a new class of neural network based density models. This new family, SNEFYs, is based on a Gaussian Process paradigm (kernel methods) and contrasts with other competitors like flow based density estimators by imposing less restrictions on the network dimensionality. SNEFYs are shown to be analytically tractable for some combinations of base measures and kernel choices, and the family of SNEFYs is moreover shown to be closed with respect to conditioning. A special case is discussed in which marginalization maintains the family as well. In a few empirical examples, SNEFYs are contrasted with Flow based and (a restricted example of) Mixture Density estimators and shown to have favorable performance.

**Strengths:**

Access to a new class of density estimators is in general a boon for the research community will very likely spur a lot of related activity.
The authors build on a framework for density estimation (fundamentally based on kernel methods) that is relatively under-explored, at least in the domain in which this reviewer is familiar with density models, which is the context of simulation based inference.
The paper is very technical, but in general quite well written.


**Weaknesses:**

The paper is written in a very technical flavor, tailored very much towards experts at the intersection of neural network and kernel methods.

A clearer discussion, with examples on how SNEFYs interface with, are similar too and contrast with other methods in the density modeling domain would have been helpful to understand better how exactly (or rather when) a non-expert can utilize SNEFYs.

Relatedly it is mentioned only shortly in the limitations that SNEFYs do not admit direct sampling (as flows may for example) and users have to resort to e.g. rejection sampling. This feature should be part of the initial discussion of other methods and the relative strengths and weakness of SNEFYs.

The disconnect between Flows and SNEFYs boil down to the difference in architecture limitations between the two methods, a priori not suggesting necessarily where in application domains the respective strengths may lie. However the examples gave the impression that SNEFYs seem specifically to outperform Flows on low dimensional problems (where flows are severely limited in widths and have to make up for in solely through depth of the transformation). Numerical examples (and figures) that clearly delineate where the performance gain emerges and where it (potentially) diminishes would have been illustrative and probably crucial cornerstones for practitioners to come away with an intuition concerning relevant application areas for SNEFYs.

**Questions:**

--

**Limitations:**

I considered the limitations section adequate, apart from a specific comment about sampling, which I suggest under "weaknesses" should have been made earlier in the paper.

---

> ### Author Rebuttal · Authors · 2023-08-04
>
> Thank you for your positive review. We are glad that you believe this area of interest (new classes of density estimators) will very likely spur a lot of related activity. We agree that this area is relatively under-explored, and are happy to hear that you found the paper well-written. We address your individual concerns below. Please let us know if you would like us to elaborate further.
>
> **Weaknesses:**
> - " *The paper is written in a very technical flavor, tailored very much towards experts at the intersection of neural network and kernel methods.  A clearer discussion, with examples on how SNEFYs interface with, are similar too and contrast with other methods in the density modeling domain would have been helpful to understand better how exactly (or rather when) a non-expert can utilize SNEFYs.* " \
> Thanks for your suggestion. We will add a paragraph to the updated manuscript. In short, SNEFYs admit closed-form normalising constants, even under marginalisation and conditioning. This stands in contrast with energy based models, which do not admit closed-form normalising constants. Normalising flows can be used to model joint or conditional distributions with closed form normalising constants. However, we are not aware of any normalising flow which can be conditioned or marginalised post-hoc in closed form. This means that one is not able to use partial observations in a (joint) density estimation task via maximum likelihood. The closest we are able to find for normalising flows that deal with missing data is an approach that jointly optimises conditional distributions for missing data and model parameters [50] or samples from missing data and optimises for model parameters. This is not maximum likelihood or sampling from a joint likelihood. In contrast, we probabilistically marginalise out missing data exactly, so that we may perform maximum likelihood. This can be beneficial when dealing with missing data --- see figure and text in global rebuttal. Further, we may use SNEFYs even without constraining the neural network mapping to be bijective. If the user suspects that the target random variable is not an invertible transformation of some simple base distribution, they might consider using a SNEFY. Having said that, normalising flows may be a better modelling choice when the user requires samples from the learnt distribution (see your next point below).
>
> - "*it is mentioned only shortly in the limitations that SNEFYs do not admit direct sampling (as flows may for example) and users have to resort to e.g. rejection sampling. This feature should be part of the initial discussion of other methods and the relative strengths and weakness of SNEFYs.*" \
> We agree with this point that this should be brought up earlier in the paper. See "Sampling" section of "General response to reviewers". In short, we will update the Introduction section of our paper to state that our method does not always allow for sampling (beyond naive rejection sampling), but normalising flows do allow this.
>
> - " *The disconnect between Flows and SNEFYs boil down to the difference in architecture limitations between the two methods, a priori not suggesting necessarily where in application domains the respective strengths may lie. However the examples gave the impression that SNEFYs seem specifically to outperform Flows on low dimensional problems (where flows are severely limited in widths and have to make up for in solely through depth of the transformation). Numerical examples (and figures) that clearly delineate where the performance gain emerges and where it (potentially) diminishes would have been illustrative and probably crucial cornerstones for practitioners to come away with an intuition concerning relevant application areas for SNEFYs.* \
> We have added one additional experiment that shows where a performance gain emerges compared with normalising flows. This is given in the general rebuttal, and concerns situations where data is missing. SNEFYs are able to be marginalised in closed-form (Theorem 2), so they can be trained using maximum likelihood estimation using partial observations. Compared with throwing away missing datapoints, SNEFYs trained using partial observations show a large increase in predictive performance. They also improve against normalising flows that discard partial observations.
>
> [50, not cited in original submission] Training Deep Normalizing Flow Models in Highly Incomplete Data Scenarios with Prior Regularization
> [51, not cited in original submission] MCFlow: Monte Carlo Flow Models for Data Imputation

---

> > ### Comment · Reviewer_tpyH · 2023-08-15
> >
> > Thank you for your clarifications. I considered this paper worth publishing a priori, however the improvements delineated will make it more readable to help especially practitioners to see the potential of this work. The emphasis on missing data scenarios is useful in distinguishing the power of flows and SNEFY's in respective applications.
> >
> > Sampling remains a weakness (now more prominently stated in the paper), but this work is a significant step forward.

---

### Author Rebuttal · Authors · 2023-08-04

**General response to reviewers:**

We thank the reviewers for their careful, detailed feedback and overall positive evaluation. We address commonly raised concerns here and respond to individual queries separately. We look forward to further engaging with the reviewers during the reviwer-author discussion period, should they have any additional concerns.

**Sampling**

We completely agree that sampling is the biggest limitation of our work. We tried to emphasise this limitation (see reviews of zjNK, k4Y8). Following tpyH's suggestion to emphasise the sampling limitation earlier on, we will update the Introduction section of our paper to state that our method does not always allow for sampling (beyond naive rejection sampling), but normalising flows do allow this. The problem we are aiming to solve is (conditional) density estimation, different to sampling. Many of the best sampling tools (GANs, VAEs, ...) do not admit likelihood calculation. Normalising flows are to the best of our knowledge unique in this respect, paying the price of requiring a bijective transformation.

We will elaborate in our future work section, which currently states that "Future work will focus on sampling, as has been done with related models [ 24 ]," so that includes a more concrete description. In particular, the current results of [24] allow for principled sampling in the special case when the activation function $\sigma$ is Gaussian and the support $\mathbb{X}$ is a hyperrectangle. It obtains $r$ approximate samples with $\mathcal{O}\big(r \log_2 (|\mathbb{X}|)) + r d \log_2\frac{2}{\rho} \big) + 1 $ evaluations of a normalising constant (which for us, costs $\mathcal{O}(m^2n + dn^2 )$). Here $\rho$ is an approximation tolerance parameter. Note that this complexity significantly improves upon naive rejection sampling, which typically scales exponentially in dimension $d$. There is nothing special about the Gaussian activation function for the algorithm to run, but the approximation error is only computed for the Gaussian activation. Also, the support $\mathbb{X}$ need not be a hyperrectangle. In short, yes, we agree that sampling is a current limitation of SNEFY. However, like other related models [23,24,38], there are bespoke sampling algorithms in special cases and future avenues for expanding the sampling capabilities of SNEFY.

**New experiment on partial observations**

We will add the following experiment. We consider the joint distribution of a random vector $\mathsf{x} \in \mathbb{R}^6$, with samples from the joint inputs and outputs of the astronomy dataset. The original training dataset is a matrix in $\mathbb{R}^{74309 \times 6}$. We partition this matrix into batches of size $256$ and randomly set each column of each batch to NaN with probability $p$ (i.e. the corresponding dimension is missing from observations). We then train a SNEFY model that utilises partial observations. We leave the original testing dataset, a matrix in $\mathbb{R}^{74557 \times 6}$, untouched, and measure the test NLL after training. We vary $p$ between $0.01$ and $1.0$ in steps of $0.01$, plotting the NLL as a function of $p$.  We also compare the performance of the SNEFY that uses partial observations with the performance of SNEFY that simply throws away incomplete observations, as well as the normalising flow baselines. Our observations are twofold: (1) Including partial observations improves performance and (2) Adding more complete observations improves performance. See attached pdf.

---

### Decision · Program_Chairs · 2023-09-21

**Decision:**

Accept (spotlight)

**Comment:**

The paper proposes a new type of density model which facilitates tractable marginals and conditionals. The paper is solid, original and of wide interest to the NeurIPS community. A clear accept, and I think this paper would be a good candidate for spotlight.

There were some minor criticisms by the reviewers, which, however, don't hamper the quality of the paper. As far as possible, the authors should please consider them for the camera-ready version.